# TRIM5α recruits HDAC1 to p50 and Sp1 and promotes H3K9 deacetylation at the HIV-1 LTR

Xiang-Hong Ran[1,4], Jia-Wu Zhu[2,4], Run-Ze Ni[1,4], Yong-Tang Zheng [3], Ya-Yun Chen[1], Wei-Hua Zheng[1] & Dan Mu[1] ✉

Tripartite motif-containing protein 5α (TRIM5α) is generally known to block the postentry events of HIV-1. Here, we report an uncharacterized role for TRIM5α in the maintenance of viral latency. Knockdown of TRIM5α potentiates the transcription of HIV-1 in multiple latency models, which is reversed by shRNA-resistant TRIM5α. TRIM5α suppresses TNFα-activated HIV-1 LTR-driven as well as NF-κB- and Sp1-driven gene expression, with the RING and B-box 2 domains being the essential determinants. Mechanistically, TRIM5α binds to and enhances the recruitment of histone deacetylase 1 (HDAC1) to NF-κB p50 and Sp1. ChIP–qPCR analyses further reveal that the association of TRIM5α with HIV-1 LTR induces HDAC1 recruitment and local H3K9 deacetylation. Conserved suppression effects of TRIM5α orthologs from multiple species on both HIV-1 and endo-retroelement HERV-K LTR activities have also been demonstrated. These findings provide new insights into the molecular mechanisms by which proviral latency is initially established and activatable proviruses are resilenced by histone deacetylase recruitment.

The coevolution of host and retroviral elements involves a fine-tuned interaction to maintain the homeostasis of host genome integrity, development and the immune system; in the absence of this homeostasis, diseases or infections can occur. Infection with human immunodeficiency virus type 1 (HIV-1) leads to the integration of the provirus into the host genome, resulting in maintained latency in a few CD4 + T cells, which is the main barrier to eradicating HIV-1 from infected individuals by current therapeutics[1,2]. HIV latency is associated with a lack of proviral gene expression, which is governed by the enhancer and promoter elements contained within the HIV-1 5' end long terminal repeat (LTR). A number of factors have been proposed to regulate the transcriptional activity of LTRs, including the local chromatin environment, the inactivation of signal-responsive transcription factors, specific transcriptional repressors and epigenetic regulatory factors that bind to the proviral LTR[3,4]. However, the mechanism of

LTR-driven expression regulation by chromatin organization and transcriptional repression is far from fully elucidated.

The latent transcriptionally repressed proviral LTR is organized into a configuration where two nucleosomes (Nuc-0 and Nuc-1) are positioned upstream of the enhancer region and immediately downstream of the transcriptional start site, respectively[4,5]. These positioned nucleosomes can undergo deacetylation and methylation, causing the provirus to achieve a heterochromatin-like state that causes transcriptional silencing. The enhancer and core promoter region, which is between Nuc-0 and Nuc-1, is accommodated by host cell factors that are critical for the activation of viral transcription, including activator protein 1 (AP-1), nuclear factor of activated T cells (NFAT), nuclear factor kappa B (NF-κB) p65–p50 heterodimers and specificity protein 1 (Sp1)[6–8]. In the presence of the NF-κB stimulus, the p65–p50 heterodimer is released from the specific inhibitor IκBα for

[1]Institute of Life Sciences, Chongqing Medical University, Chongqing, China. [2]School of Basic Medical Sciences, Kunming Medical University, Kunming, Yunnan, China. [3]Key Laboratory of Animal Models and Human Disease Mechanisms of the Chinese Academy of Sciences, Kunming Institute of Zoology, Chinese Academy of Sciences, Kunming, Yunnan, China. [4]These authors contributed equally: Xiang-Hong Ran, Jia-Wu Zhu, Run-Ze Ni. ✉ e-mail: danmu@cqmu.edu.cn

translocation to the nucleus, where it binds to HIV LTR to activate transcription[7]. In unstimulated cells, p50–p50 homodimers are the predominant nuclear species of NF-κB proteins that bind to the LTR and recruit histone deacetylases (HDACs), which leads to deacetylation of surrounding histones and transcription silencing[9]. Sp1 is also a target for HDAC-mediated transcriptional repression[8,10]. Several studies have reported that HDACs are recruited to Sp1 by assistant cofactors, such as c-Myc and COUP-TF interacting protein 2 (CTIP2)[8,11]. These findings highlight that cellular factors recruiting HDACs to maintain the quiescence of the HIV-1 LTR are important targets for anti-latency therapies.

TRIM5α, an ancient and ubiquitously expressed member of the tripartite motif-containing (TRIM) family, was initially documented as a restriction factor against a range of viruses, including retroviruses, foamy viruses and flaviviruses[12–15]. TRIM5α comprises RING, B-box 2, and coiled-coil domains that are common to all TRIM proteins and a C-terminal PRY-SPRY domain[16]. The coiled-coil domain is important for TRIM5α dimerization, and the B-box 2 domain mediates higher-order self-association[17–19]. The RING domain of TRIM5α confers E3 ligase activity mediating ubiquitination and is also implicated in promoting the efficiency of higher-order assembly together with the B-box 2 domain[17]. The C-terminal PRY-SPRY domain is responsible for recognizing and binding to the capsid of retroviruses[20]. TRIM5α and its alternatively spliced isoform TRIM5-CyclophilinA (TRIMCyp) can block the reverse transcription stage of HIV-1 replication in a species-specific manner, with macaque orthologs potently inhibiting HIV-1 but human TRIM5α (huTRIM5α) weakly restricting HIV-1[21–24]. However, accumulating evidence has shown that huTRIM5α is able to block HIV-1 replication in Langerhans cells, CD4 + T cells and dendritic cells via various mechanisms[25–27]. While these processes are observed in the cytoplasmic compartments, TRIM5α has also been reported to be present in the nucleus by a series of studies. A study showed that TRIM5α accumulated in the nucleus by the use of LMB, a drug that blocks proteins from translocating into the cytoplasm[28]. Another study demonstrated that TRIM5α was sequestered in human and rhesus macaque dendritic cell nuclei, leading to efficient cellular innate sensing[29]. However, whether TRIM5α participates in any other processes that take place in the nucleus remains unexplored.

Here, we report that TRIM5α inhibits HIV-1 gene transcription by promoting the recruitment of HDAC1 to p50 and Sp1, consequently facilitating the formation of a heterochromatic environment at the HIV-1 LTR and leading to HIV-1 latency. Knockdown of TRIM5α potentiates the basal and tumor necrosis factor alpha (TNFα)-induced transcription of HIV-1 in multiple latency models, which is reversed by an shRNA-resistant TRIM5α mutant. Mechanistically, TRIM5α suppresses the TNFα-stimulated transcriptional activity of HIV-1 LTR by downregulating both Sp1- and NF-κB-driven gene expression, with the RING and B-box 2 domains being the essential determinants. Coimmunoprecipitation assays show that TRIM5α binds to and strongly enhances the formation of p50–HDAC1 and Sp1–HDAC1 complexes via the RING and B-box 2 domains. Chromatin immunoprecipitation (ChIP)–qPCR analysis further demonstrates that TRIM5α is recruited to and promotes the binding of HDAC1 to the HIV-1 LTR promoter, leading to local histone deacetylation. Furthermore, TRIM5α orthologs from human and nonhuman primates show conserved suppression effects on the LTRs of HIV-1 and human endogenous retrovirus K101 (HERV-K101). Overall, these findings identify TRIM5α as a potential transcription corepressor facilitating retroviral silencing, providing new therapeutic opportunities.

## Results

### Knockdown of endogenous human TRIM5α reactivates HIV-1 latency in cellular models

To investigate whether endogenous TRIM5α suppresses HIV-1 gene expression in CD4 + T cells, we first performed experiments in chronically HIV-1-infected Jurkat CD4 + T cells (J-Lat) harboring a full-length HIV provirus with a green fluorescent protein (GFP) gene in place of the nef gene, thus providing a cellular model for HIV-1 latency (Supplementary Fig. 1a). TRIM5α-knockdown (TRIM5α-KD) J-Lat cells were generated by transducing lentiviral particles encoding shRNA targeting TRIM5α, with cells transduced with scramble shRNA lentivirus as a negative control and TRIM28-knockdown (TRIM28-KD) cells as a positive control. Strongly reduced endogenous expression levels of TRIM5α or TRIM28 were observed compared to those in the scramble cells by both qPCR and Western blot (Fig. 1a, b). As reported in previous studies, knockdown of TRIM28 led to reactivation of HIV-1 in J-Lat cells[30,31]. The TRIM5α-KD J-Lat cells showed similar reactivation levels of HIV-1 latency compared with TRIM28-KD cells in the absence or presence of TNFα (Fig. 1c), one of the latency reversing agents (LRAs) that are involved in NF-κB pathway-dependent HIV-1 transcription stimulation[32]. Furthermore, we showed that TRIM5α suppressed HIV-1 latency reactivation in the presence of different dosages of TNFα treatment (Supplementary Fig. 1b). Similar to TNFα, phorbol 12-myristate 13-acetate (PMA) and prostratin, which have also been reported to induce HIV-1 expression by NF-κB and Sp1 activation[33,34], caused a ~200% enhancement of LTR-driven GFP reporter expression in the TRIM5α-KD J-Lat cells compared with the scramble control cells (Supplementary Fig. 1b). However, treatment with JQ1, a small molecular inhibitor of BET bromodomains that stimulate Pol II elongation[35], failed to induce enhanced reactivation of HIV-1 latency in the TRIM5α-KD cells compared to the control cells (Supplementary Fig. 1b), suggesting that the effect of JQ1-mediated activation saturates the effect of TRIM5α. Moreover, the mRNA levels of HIV-1 Gag and Tat-Rev in J-Lat cells were elevated by the knockdown of endogenous TRIM5α in either the absence or presence of TNFα (Fig. 1d).

To verify that the HIV-1 latency reversal displayed in J-Lat cells was due to the absence of TRIM5α, an shRNA-resistant TRIM5α-KD J-Lat cell line with a pCDH-vector-based nontargeting TRIM5α mutant (ntTRIM5α) was reconstituted. The immunoblots showed that the ntTRIM5α construct increased TRIM5α expression in the scramble cells (lane 2, Fig. 1e) and reversed the expression level of TRIM5α in the TRIM5α-KD cells (lane 4, Fig. 1e), as verified by both HA-specific and TRIM5α-specific antibodies. As Fig. 1f shows, compared to cells transduced with empty lentivectors, scramble+ntTRIM5α cells exhibited reduced basal LTR-driven GFP transcription (7.26% vs. 4.82%) and impaired latency reactivation by TNFα (26.1% vs. 19.4%), demonstrating that the ectopic expression of TRIM5α has a downregulatory effect on HIV-1 LTR-driven transcription. Notably, TRIM5α-KD+ntTRIM5α J-Lat cells lost the ability to enhance basal and TNFα-reactivated GFP expression compared to TRIM5α-KD+pCDH J-Lat cells (basal: 8.16% vs. 11.3%; reactivated: 29.8% vs. 42.7%), showing an HIV-1 latency state similar to that in the scramble+pCDH cells (Fig. 1f).

We further investigated the effect of TRIM5α knockdown on the reactivation of a HeLa-derived TZM-bl cell line, which contains HIV-1 integrated with an LTR-driven luciferase (luc)-reporter. The TRIM5α knockdown efficiency was confirmed by qPCR and Western blot analysis (Fig. 2a, b). The reduction in endogenous TRIM5α increased both the basal and reactivated luciferase activity after treatment with TNFα, PMA and prostratin but not JQ1 (Fig. 2c), consistent with the results from J-Lat cells (Supplementary Fig. 1b). Furthermore, we established a primary CD4 + T HIV-1 latency model by infecting cells with a luciferase-reporter pseudotyped HIV-1 vector, followed by transduction with shRNA-TRIM5α or shRNA-scramble lentivectors. Successful infection at 2 days postinfection (dpi) and establishment of latent status at 7 dpi and 10 dpi of HIV-1 were confirmed by measuring luciferase activity (Fig. 2d). Cells were transduced with shRNA-scramble or shRNA-TRIM5α lentiviruses at 7 dpi, followed by puromycin treatment at 10 dpi. The efficient knockdown of TRIM5α in each donor was shown by qPCR analysis (Fig. 2e). Latent HIV-1 was reactivated by CD3/CD28 antibody Dynabeads at 13 dpi. Three days post reactivation, the

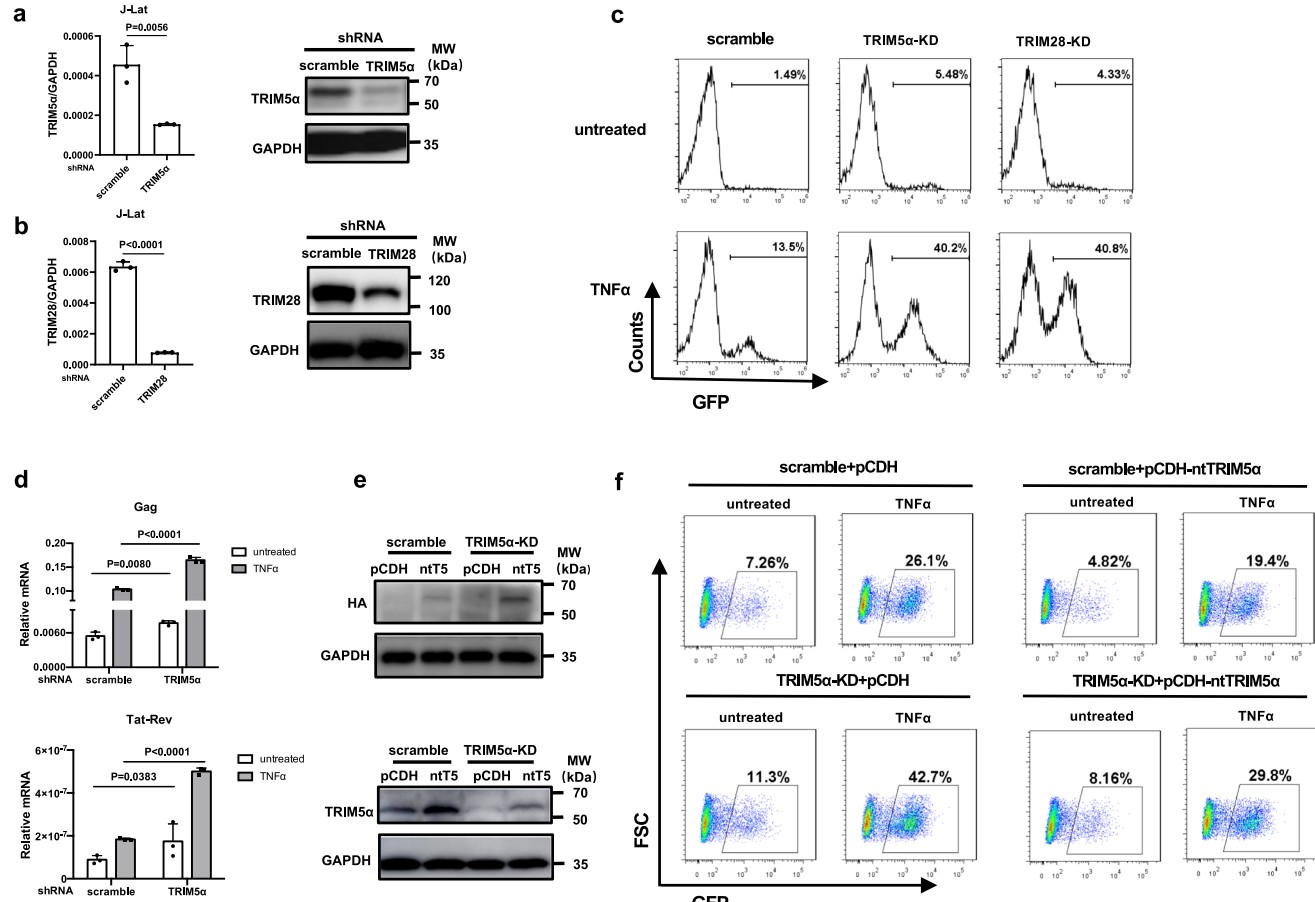

**Fig. 1 | Knockdown of endogenous TRIM5α reactivates HIV-1 latency in J-Lat CD4 + T cells. a**, **b** J-Lat CD4 + T cells were stably transduced with shRNA targeting TRIM5α (**a**) or TRIM28 (**b**), with cells transduced with scrambled shRNA as control. The knockdown effect of endogenous TRIM5α or TRIM28 was assessed by qPCR analysis and Western blot. **c** TRIM5α-knockdown (KD), TRIM28-KD or scramble-KD J-Lat were stimulated with or without TNFα (10 ng/ml). The reactivation of HIV-1 transcription was measured by detecting GFP expression via flow cytometry. **d** TRIM5α-KD or scramble-KD J-Lat cells were stimulated with or without TNFα (20 ng/ml) for 24 h before the Gag and Tat-Rev mRNA expression was examined via

qPCR. **e** Expression of TRIM5α in the TRIM5α-KD J-Lat by transducing lentivectors carrying an shRNA-resistant TRIM5α-HA construct. The protein levels were detected by both anti-HA and anti-TRIM5α antibodies. **f** J-Lat cells stably transduced with shRNA-scramble+pCDH, shRNA-scramble+pCDH-nontargetingTRIM5α (ntT5) -HA, shRNA-TRIM5α + pCDH, shRNA-TRIM5α + pCDH-ntT5-HA were stimulated with or without TNFα for 24 h. GFP expression in each group was examined to compare the HIV-1 latency reversal. Data represent at least 3 independent experiments and are presented as mean ± SD (*n* = 3). The statistical significance analyses were performed using a two-sided unpaired t test. Source data are provided as a Source data file.

luciferase activity of primary CD4 + T cells expressing reduced TRIM5α was significantly enhanced compared to that of the scramble control cells (Fig. 2f).

Collectively, these data show that endogenous TRIM5α has the ability to prevent HIV-1 gene expression under basal conditions or in the presence of selected LRAs.

## TRIM5α suppresses the reactivation of HIV-1 LTR-driven gene expression

It has been shown that transcriptional silencing of HIV-1 replication is associated with multiple layers of regulation, including inadequate availability of transcription factors, recruitment of transcription repressors, and epigenetic modifications at the HIV-1 LTR[3,4]. We then set out to examine whether TRIM5α exerts a direct effect on the HIV-1 LTR by employing an LTR-driven firefly luciferase reporter assay. The luciferase activity of the reporters was normalized to the luciferase activity of RL-TK (*Renilla* luciferase). A full-length HIV-1 LTR that encompasses the major transcription factor regulatory elements and Tat-responsive TAR was first utilized (Fig. 3a). Overexpression of TRIM5α caused ~50% suppression of TNFα-induced HIV-LTR-full activity, with no suppression of Tat-induced LTR-full-luc activity observed (Fig. 3b, c). In contrast, the basal full-length HIV-1 LTR activity

was not influenced by TRIM5α overexpression (Fig. 3c). Next, to determine whether TRIM5α regulates HIV-1 LTR by exerting an effect on NF-κB and Sp-1, we constructed a truncated LTR (hereafter referred to as "core LTR") that only harbored two NF-κB binding sites, three Sp-1 binding sites, and a TATA box (Fig. 3a). Again, overexpression of TRIM5α resulted in ~50% suppression of TNFα-induced HIV-LTR-core luciferase reporter activity compared to the empty vector control, while the repression of the basal transcriptional activity of the core LTR was minimal (Fig. 3d). In addition, TRIM5α repressed TNFα-induced LTR activation in a dose-dependent manner (Fig. 3e).

Next, we engineered a stable TRIM5α-KD 293 T cell line for the LTR-driven luciferase reporter assay (Fig. 3f, g). Compared to the scramble-shRNA-transduced 293 T cells, the TRIM5α-KD 293 T cells displayed enhanced luciferase activity for both LTR-full- and LTR-core-driven transcription in the presence of TNFα (Fig. 3h, i). We also employed the CRISPR–Cas9 technique to generate 293T-TRIM5α-knockout (KO) single clone cell lines. The sgRNA target is shown in Fig. 3j. Successful establishment of two TRIM5α-deficient mono-clonal cell lines was verified by Western blot and Sanger sequencing (Fig. 3k, l). Again, LTR-full- and LTR-core-driven transcription activities were more efficiently activated by TNFα in the TRIM5α-KO 293 T cells than in the control cells (Fig. 3m, n). Taken together, these data

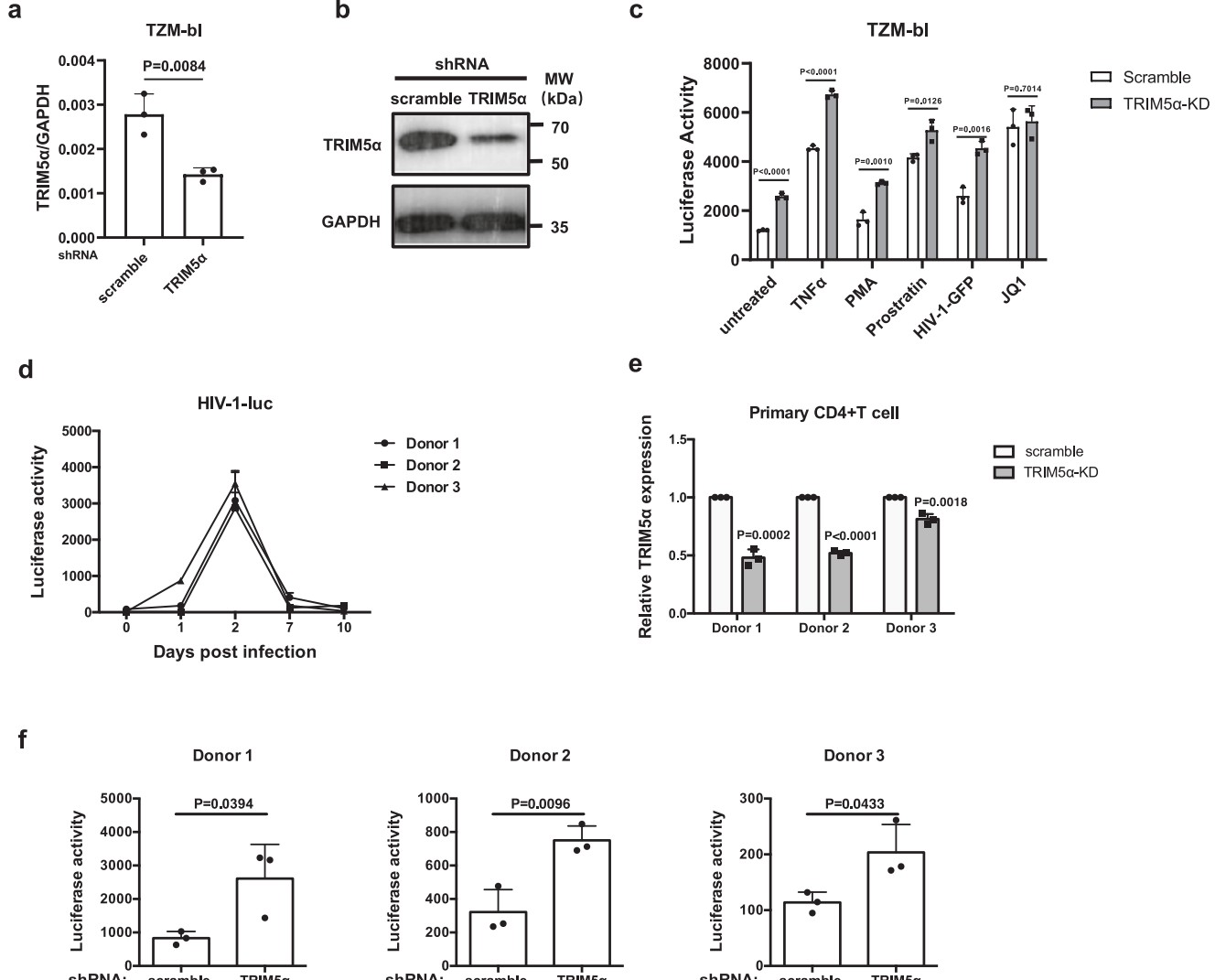

**Fig. 2 | Knockdown of TRIM5α in TZM-bl and primary CD4 + T cells reactivates HIV-1 transcription. a, b** qPCR analysis (**a**) and Western blot (**b**) confirmed the efficient knockdown of TRIM5α in TZM-bl cells. **c** TRIM5α-knockdown (KD) or scramble-KD TZM-bl cells were seeded in 96-well plates and then treated with various LRAs including TNFα (20 ng/ml), PMA (100 nM), prostratin (2.5 μM), HIV-1-GFP (50 μl), or JQ1 (2.5 μM) for 24 h. The luciferase activity of cell lysates was detected to indicate HIV-1 transcription in the TZM-bl cells. Data represent at least 3 independent experiments and are presented as mean ± SD (*n* = 3). The statistical significance analyses were performed using a two-sided unpaired *t* test. **d** Primary CD4 + T cells derived from 3 healthy donors were activated by anti-CD3/CD28 antibody-coated beads for 3 days and were then infected with a single-cycle HIV-1

pNL4-3 containing a luc reporter pseudotyped with VSV-G to produce a primary T cell model of latency. Cells were collected on day 1 (post-infection day of pNL4-3-luc), day 2, day 7, and day 10 to follow the development of the latency model. **e** At day 7 dpi, latent primary CD4 + T cells were transduced with shRNA-scramble or shRNA-TRIM5α lentivirus for 3 days and then selected by puromycin for another 3 days. mRNA expression level of TRIM5α in HIV-1 latently infected primary CD4 + T cells transduced with shRNA-scramble or shRNA-TRIM5α lentivirus. **f** To reactivate HIV-1 transcription, anti-CD3/CD28 antibody-coated beads were added to cells for a 3-day stimulation before the luciferase activity was measured. Data represent mean ± SD (*n* = 3). The statistical significance analyses were performed using a two-sided unpaired t test. Source data are provided as a Source data file.

suggest that human TRIM5α suppresses TNFα-induced HIV-1 LTR-driven gene expression.

## Mapping the determinants of HIV-1 LTR that are involved in TRIM5α-mediated LTR regulation

To map the specific determinants in the core LTR that may serve as targets of TRIM5α-mediated LTR suppression, we constructed two mutant HIV-LTR-core reporter plasmids by deleting the NF-κB- and Sp1-binding sites. Consistent with previous reports[36,37], TNFα marginally induced the activity of the ΔNF-κB core LTR, which contains only three Sp1 binding sites, by 2-fold (Fig. 4a), while the ΔSp1 core LTR, which harbors two κB sites, showed ~100-fold induction upon TNFα treatment (Fig. 4b). The suppressive effect of TRIM5α remained on both the ΔNF-κB and ΔSp1 core LTRs under TNFα treatment (Fig. 4a,

b), suggesting that the remnant regulatory element of each LTR mutant, that is, the Sp1 and κB sites, respectively, remained a target of TRIM5α. However, in the absence of TNFα treatment, TRIM5α suppressed the transcriptional activity of the ΔNF-κB core LTR while activating the ΔSp1 core LTR (Fig. 4a, b), presumably because TRIM5α stimulates NF-κB-luc activity[38]. Thus, these data indicate a differential modulatory effect of TRIM5α on the Sp1 and NF-κB elements of the core LTR under basal conditions.

To further define the specific target sites, we examined the direct impact of TRIM5α on the individual NF-κB or Sp1 binding sites by a 5*κB or a 3*Sp1 luciferase reporter gene, respectively. Consistent with previous reports[38,39] and the result from the ΔSp1 core LTR reporter assay (Fig. 4b), TRIM5α activated the 5*κB reporter in the absence of a stimulus (Fig. 4c). In contrast, the induction of 5*κB reporter activity by

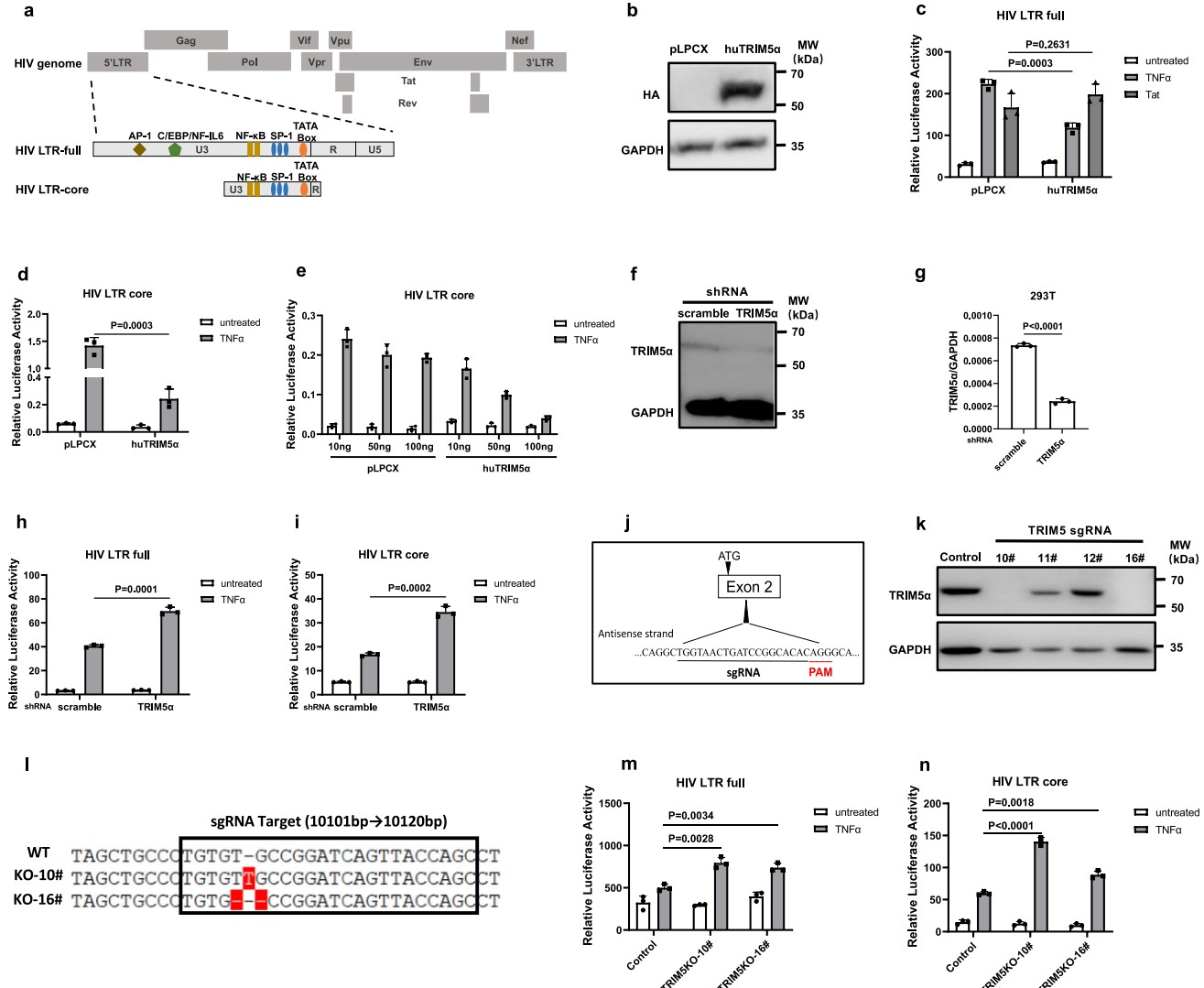

**Fig. 3 | TRIM5α suppresses the reactivation of HIV-1 LTR-driven gene expression in 293 T cells. a** Schematic diagram of the full-length and the core elements of HIV-1 LTR that were used for constructing the luciferase reporters. **b** Immunoblot analysis of transient overexpression of human TRIM5α in 293 T cells. **c** TRIM5α (100 ng) or pLPCX vector (100 ng) was cotransfected into 293 T cells with pGL-HIV-LTR-full-luc construct (100 ng). pRL-TK plasmid (20 ng) was transfected along for normalization of transfection efficiency. Twenty-four hours post-transfection, cells were treated with or without TNFα (20 ng/ml) or the 2nd transduction of pcDNA3.1-Tat. Cells were harvested in 24 h and measured for luciferase activity. **d** TRIM5α (100 ng) or pLPCX (100 ng), pGL-HIV-LTR-core-luc (100 ng) and pRL-TK was cotransfected in 293 T cells. Twenty-four hours post-transfection, cells were treated with or without TNFα (20 ng/ml) for 24 h before luciferase activity was examined. **e** A dose-dependent assay was performed to detect the effect of TRIM5α on the HIV-LTR-core-driven transcription. **f, g** Immunoblot analysis of TRIM5α protein expression (**f**) and mRNA expression (**g**) of stable TRIM5α-knockdown or control

293 T cells. **h, i** TRIM5α-knockdown or control cells were seeded and transfected with pGL-HIV-LTR-full-luc (100 ng) (**h**) or pGL-HIV-LTR-core-luc (100 ng) (**i**), along with pRL-TK. Twenty-four hours post-transfection, cells were treated with or without TNFα (20 ng/ml). The luciferase activity was measured at 24-h post-treatment. **j** CRISPR–Cas9 technique was used to generate TRIM5α-knockout (KO) single clone cell lines. The sgRNA target was shown. **k** Multiple single clone cell lines were subjected to Western blot by an anti-TRIM5α antibody. **l** TRIM5α-KO clone 10# and 16# was verified by Sanger sequencing. **m, n** TRIM5α-KO clone 10#, 16# or control 293 T cells were transfected with pGL-HIV-LTR-full-luc (100 ng) (**m**) or pGL-HIV-LTR-core-luc (100 ng) (**n**), along with pRL-TK. Cells were treated with or without TNFα before the luciferase activity measurement. Data represent mean ± SD in triplicates (*n* = 3) of at least 3 independent experiments. The statistical significance analyses were performed using a two-sided unpaired *t* test. Source data are provided as a Source data file.

TNFα decreased drastically in TRIM5α-overexpressing cells compared to control cells (Fig. 4c). On the other hand, TRIM5α had a moderate impact on Sp1 reporter activity under basal conditions but showed a potent suppressive effect in the presence of TNFα stimulation (Fig. 4d). In line with these findings, the reporter activities of both NF-κB and Sp1 were significantly elevated in the TNFα-treated TRIM5α-KO 293 T cells compared with the control cells (Fig. 4e, f). Thus, these results indicate that TRIM5α exerts a suppressive effect on the HIV LTR by targeting both NF-κB and Sp1 in a latency-activated state.

The canonical activation of NF-κB involves IKK activation and degradation of IκBα, resulting in NF-κB p65 subunit translocation to

the nucleus, where it binds to and activates *cis*-acting regulatory elements on the genome[32]. We then examined whether TRIM5α knockdown augments the nuclear translocation of NF-κB upon TNFα treatment. As expected, the nuclear p65 level was elevated upon TNFα stimulation. p65 nuclear translocation in TRIM5α-KD J-Lat cells showed a decrease compared to that in control cells in the absence of TNFα but showed no difference in the presence of TNFα (Fig. 4g). Additionally, the protein levels of p65 and Sp1 were not influenced by TRIM5α knockdown (Fig. 4g and Supplementary Fig. 2a); the mRNA and protein levels of TRIM5α were not altered by TNFα treatment (Fig. 4g and Supplementary Fig. 2b, c). In agreement with previous reports showing

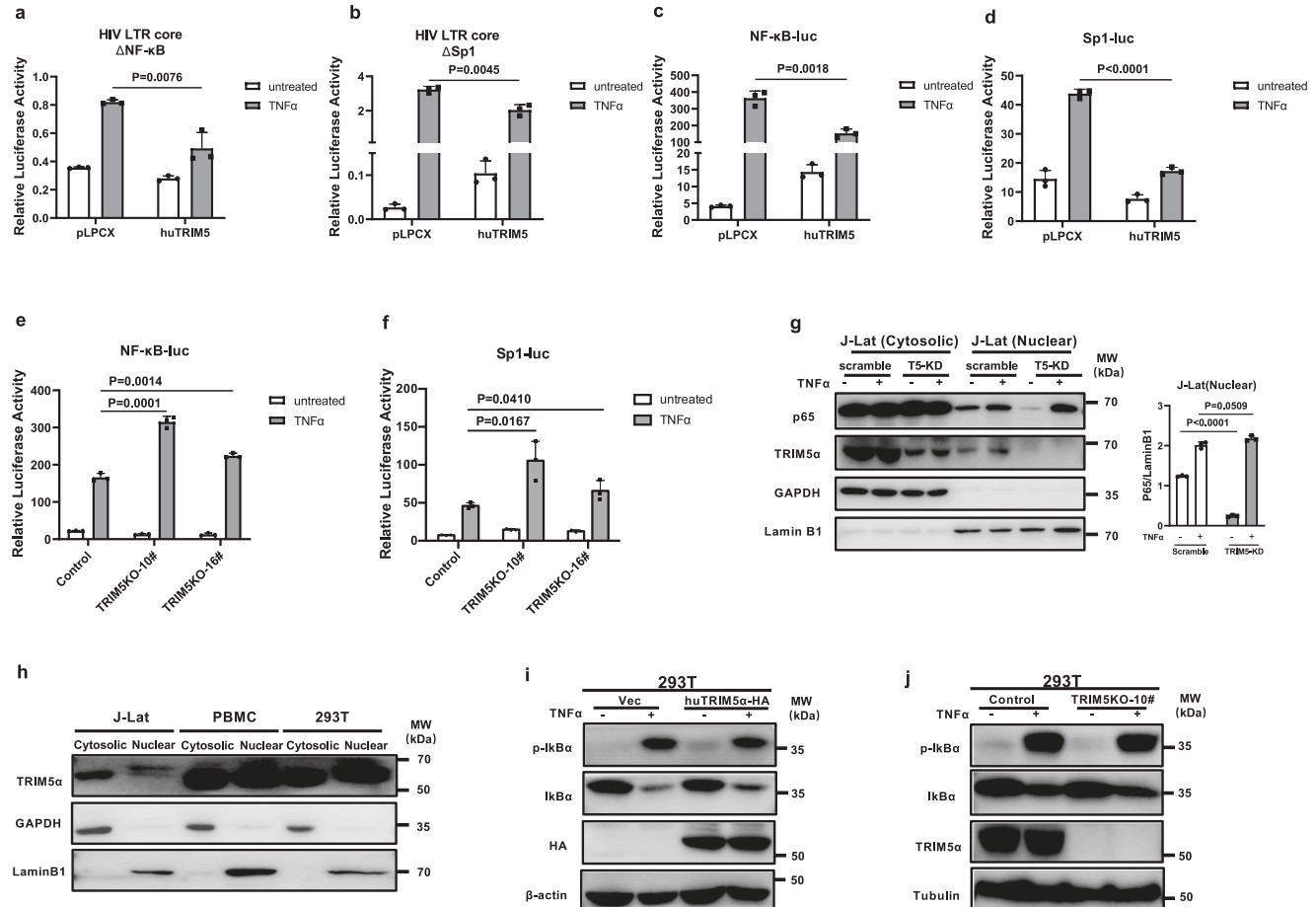

**Fig. 4 | Mapping the determinants of HIV-1 LTR involved in the TRIM5α-mediated LTR suppression. a, b** Human TRIM5α (100 ng) or pLPCX vector (100 ng) was cotransfected with mutant pGL-HIV-LTR-core-luc (100 ng) which lacks the NF-κB (**a**) or Sp1 (**b**) responsive elements. TNFα (20 ng/ml) was used to stimulate the mutant LTRs. The luciferase activity of the mutated HIV-LTR-core was normalized by the coexpressing RL-TK. **c, d** pGL vectors constructed to harbor 5*κB (**c**) or 3*Sp1 (**d**) binding sites were used to detect the effect of TRIM5α-mediated regulation. TNFα (20 ng/ml) was used as the stimulation. **e, f** TNFα-induced NF-κB-luc (**e**) or Sp1-luc (**f**) reporter activity was assessed in TRIM5α-knockout (KO) clone 10# 293 T cells. Data in **a**–**f** are representative of at least 3 independent experiments and are mean ± SD (*n* = 3). The statistical significance analyses were performed using a two-sided unpaired *t* test. **g** Nuclear transport of activated p65 was compared between TRIM5α-knockdown (KD) and scramble-KD J-Lat cells, in the absence or presence of TNFα (20 ng/ml) for 24 h. Immunoblotting was performed by a rabbit anti-p65 antibody. The relative nuclear p65 levels were compared by calculating the grayscale values of the blot bands. Endogenous expression of TRIM5α was also shown by an anti-TRIM5α antibody. Data represents mean ± SEM (*n* = 3). The statistical significance analyses were performed using a two-sided unpaired *t* test. **h** Expression of TRIM5α was detected in the cytoplasm and the nuclear extract of untransduced J-Lat cells, 293 T cells and human PBMCs by an anti-TRIM5α antibody. **i, j** Phosphorylation and degradation of IκBα induced by TNFα (20 ng/ml) for 1 h were detected in TRIM5α-overexpressing (**i**) or TRIM5α-KO (**j**) cells. Immunoblotting was performed by an anti-pIκBα and an anti-IκBα antibody. Data in **h**–**j** are representative of at least 2 independent experiments. Source data are provided as a Source data file.

that TRIM5α could be detected in the nucleus as well as cytoplasm[28,29,40,41], we observed here that TRIM5α was detectable in the nucleus in the scramble-transduced J-Lat cells by Western blot (Fig. 4g). We also detected the protein distribution of TRIM5α in untransduced J-Lat cells, 293 T cells and human PBMCs. The results showed that TRIM5α existed not only in the cytoplasm but also in the nuclear extract of all the cell types tested (Fig. 4h); confocal immunofluorescence microscopy also revealed the nuclear distribution of TRIM5α in 293 T cells (Supplementary Fig. 3). Notably, the nuclear extract of PBMCs harbored substantial amounts of TRIM5α, suggesting that TRIM5α may play an important role in PBMCs.

In addition, the effect of TRIM5α on the phosphorylation and degradation of IκBα was examined. Treatment with TNFα increased the phosphorylation and degradation of IκBα, as expected (Fig. 4i). Overexpression or knockout of TRIM5α had no or little effect on the TNFα-mediated activation of IκBα (Fig. 4i, j). Thus, these data demonstrate that TRIM5α interferes with the activity of NF-κB but has no effect on its cytosolic signaling pathway activation, suggesting that

TRIM5α plays a more direct role at the NF-κB regulatory element of the HIV-1 LTR in the nucleus.

## TRIM5α-mediated LTR regulation is dependent on the RING and B-box 2 domains

To map the determinants of the TRIM5α protein that contribute to LTR-driven transcription suppression, a series of mutants were constructed by deleting each of the four domains (Fig. 5a). The expression of each mutant was confirmed by Western blot (Fig. 5b). The promoter-reporter assays showed that deletion of the RING domain or B-box 2 domain resulted in a significant decrease in TRIM5α-mediated LTR suppression, while deletion of neither of the other two domains conferred the same effect (Fig. 5c). Similar results were observed when the suppressive activities of the TRIM5α mutants were examined on the TNFα-stimulated NF-κB-luc and Sp1-luc reporters (Fig. 5d, e). Thus, these results indicate that the RING and B-box 2 domains play a critical role in TRIM5α-mediated HIV-1 LTR suppression by interrupting NF-κB- and Sp1-driven transcription.

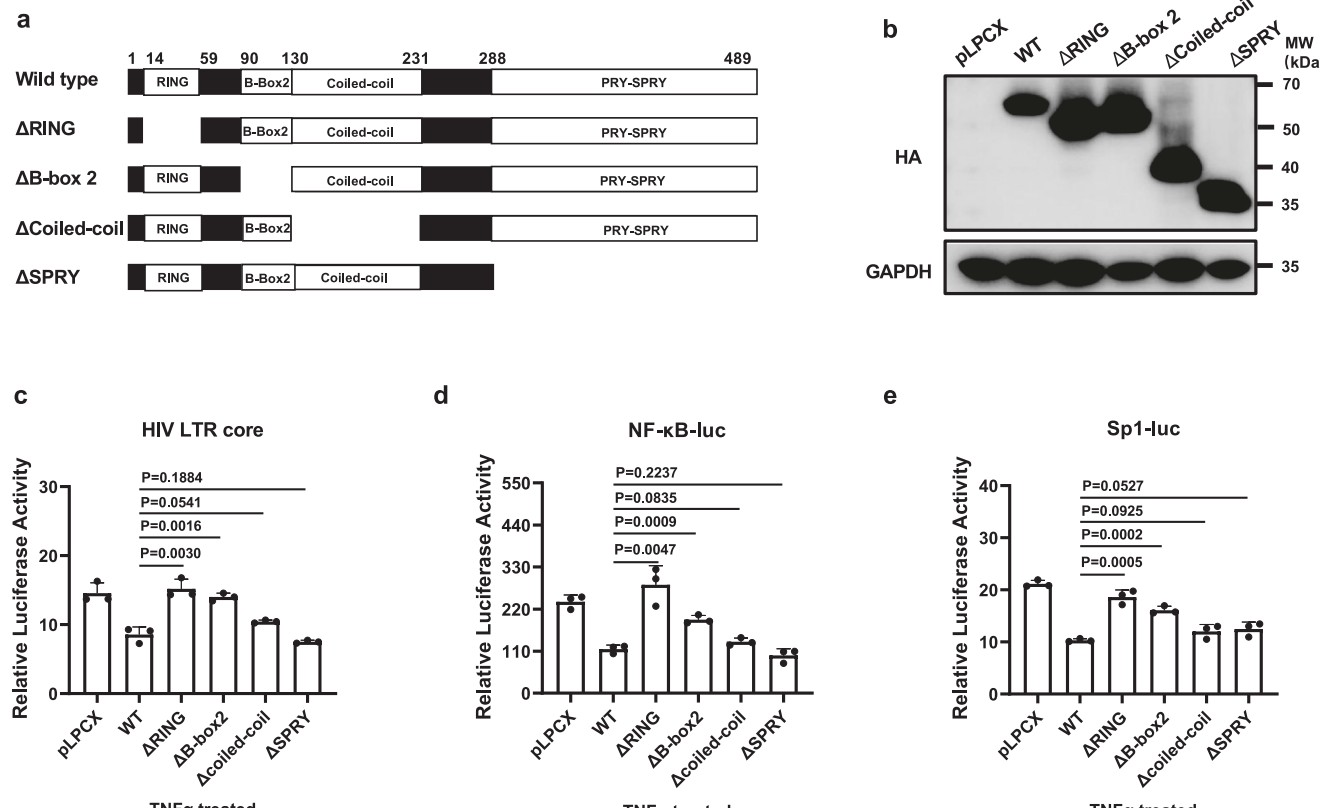

**Fig. 5 | Mapping the determinants of TRIM5α involved in the TRIM5α-mediated LTR regulation. a** Schematic diagram of the domain-deletion mutants of human TRIM5α. **b** Overexpression of HA-tagged TRIM5α mutants in 293 T was confirmed by Western blot analysis. **c–e** Suppression effect of the domain-deletion TRIM5α mutants on TNFα-driven HIV-LTR-core-luc (**c**), NF-κB-luc (**d**), and Sp1-luc (**e**) reporter activities was assessed. The luciferase activity of the reporters was normalized by RL-TK. Data are representative of at least 3 independent experiments and are mean ± SD ($n = 3$). The statistical significance analyses were performed using a two-sided unpaired $t$ test. Source data are provided as a Source data file.

## TRIM5α promotes the recruitment of HDAC1 to both Sp1 and p50 via the RING and B-box 2 domains

As TRIM5α did not suppress the activation of the NF-κB cytosolic signaling pathway, we then examined whether TRIM5α regulates the NF-κB or Sp1 regulatory element directly at the HIV-1 LTR region. NF-κB p65/p50 and Sp1 have been reported to recruit HDAC1 to the HIV-1 LTR, but the underlying mechanisms have not been fully elucidated. By using coimmunoblot assays, we showed that HDAC1 bound to Sp1, p65, and p50 (Fig. 6a–c), as previously reported[8,42,43]. We also observed that while TRIM5α had a weak association with p65 (Fig. 6c), it strongly interacted with Sp1 and p50 (Fig. 6a, b). Notably, TRIM5α greatly enhanced the recruitment of HDAC1 to both Sp1 and p50 but not p65 (Fig. 6a–c), suggesting that TRIM5α participates as a cosuppressor in the transcription machinery complex.

Next, we asked whether TRIM5α promotes the recruitment of HDAC1 to Sp1 and p50 via the RING and B-box 2 domains, which contribute to TRIM5α-mediated LTR suppression. The results showed that, compared with wild-type TRIM5α, which potently promoted the interaction of HDAC1 and Sp1 or HDAC1 and p50, the RING or B-box 2 domain deletion mutant completely abolished the HDAC1 recruitment activity (Fig. 6d, e). The coimmunoblots also revealed that both TRIM5α-ΔRING and TRIM5α-ΔB-box 2 exhibited much weaker interactions with Sp1 and p50, respectively (Fig. 6d, e). Notably, the results showed that the TRIM5α-promoted HDAC1–Sp1 and HDAC1–p50 interactions were attenuated by treatment with TNFα (Fig. 6f, g) and that the association of TRIM5α and Sp1 was also markedly reduced upon TNFα treatment (Fig. 6f), suggesting the release of TRIM5α-mediated suppression of Sp1 and NF-κB under latency-reversing stimulation.

We further examined the association between HDAC1 and wild-type or mutant TRIM5α. The results showed that TRIM5α bound to HDAC1 and that deletion of either the RING or B-box 2 domain impaired the interaction of HDAC1 and TRIM5α (Fig. 6h). As the RING domain exhibits E3 ubiquitin ligase activity and the B-box 2 domain of TRIM5α has been reported to mediate self-association[18,19], we next tried to determine whether interference with E3 ligase or higher order assembly would abolish the TRIM5α-mediated interaction of HDAC1 and Sp1 and the interaction of HDAC1 and p50. Single point mutants RING L19R, which selectively disables E2 binding and thus RING activity[44], and B-box R119E, which completely abolishes higher-order assembly[18,45], were employed.

The co-IP assay results showed that TRIM5α-R119E significantly impaired the interaction of HDAC1 and TRIM5α-WT, as well as the TRIM5α-mediated HDAC1–Sp1 and HDAC1–p50 interactions, while TRIM5α-L19R was not able to do so (Fig. 7a–c). We also observed that, compared with TRIM5α-WT or L19R, R119E exhibited a much weaker interaction with Sp1 as well as with p50 (Fig. 7b, c). Moreover, luciferase assays showed that TRIM5α R119E but not L19R greatly reduced the TRIM5α-mediated suppression of LTR, NF-κB, and Sp1 activation upon TNFα treatment (Fig. 7d). Altogether, these findings reveal that TRIM5α functions as a mediator to promote the recruitment of HDAC1 to Sp1 and p50, which is dependent on the HDAC1-interacting RING and B-box 2 domains, with R119 in the B-box 2 domain being the key determinant amino acid.

## Association of TRIM5α with the HIV-1 LTR promoter induces recruitment of HDAC1 and local histone H3 deacetylation

To strengthen the hypothesis that TRIM5α plays a direct role in the repression of HIV LTR expression, ChIP–qPCR assays were employed

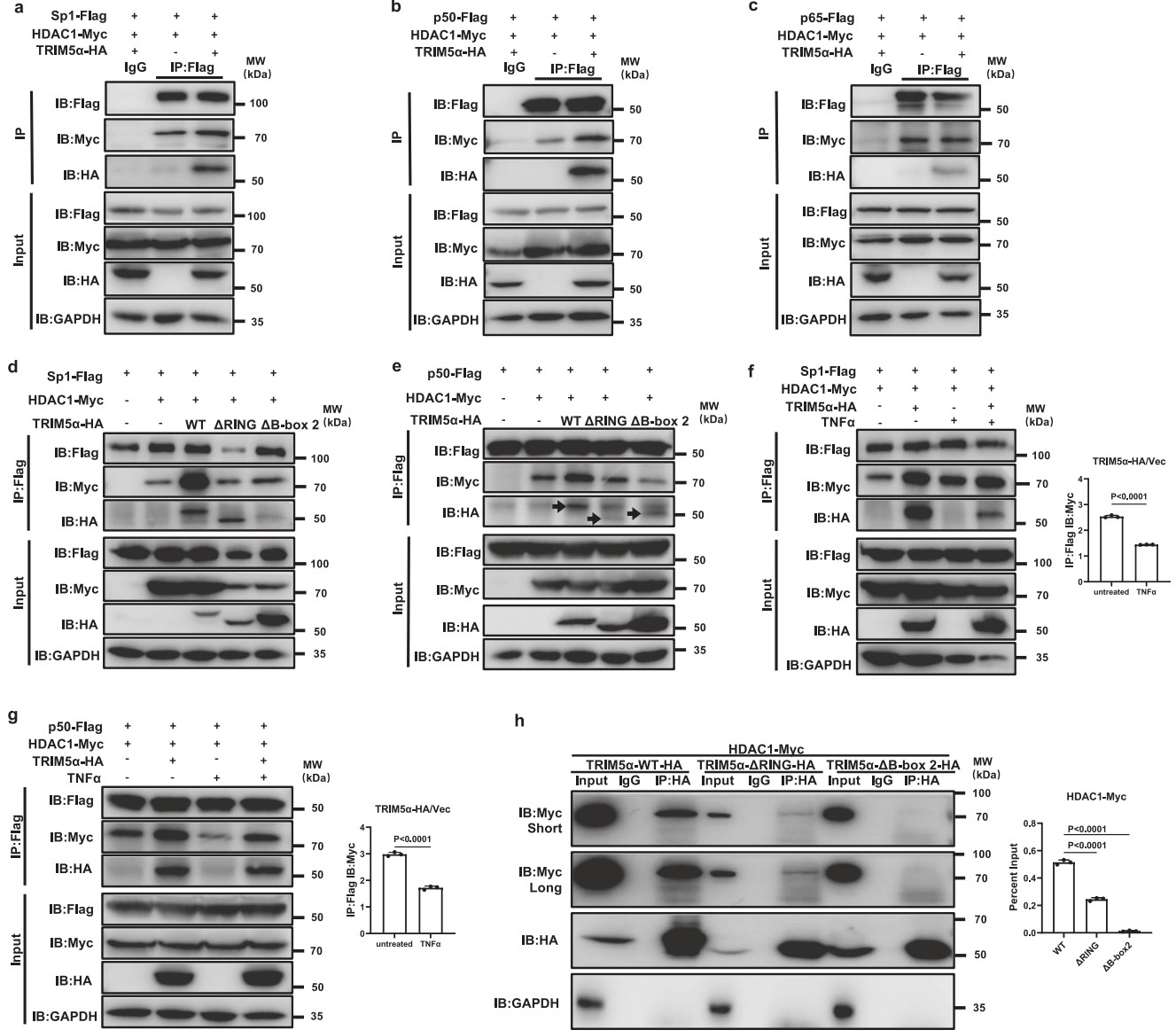

**Fig. 6 | TRIM5α promotes the formation of Sp1–HDAC1 and p50–HDAC1 complexes via the RING and B-box2 domains. a–c** Myc-tagged HDAC1 and Flag-tagged Sp1 (**a**), p65 (**b**), or p50 (**c**) were cotransfected into 293 T cells with or without HA-tagged TRIM5α. Forty-eight hours post-transfection, cells were lysed and immunoprecipitated with a mouse anti-Flag antibody. Western blot was performed with rabbit anti-Flag, anti-Myc, and anti-HA antibodies. **d, e** WT or mutant TRIM5α plasmid was cotransfected with Myc-tagged HDAC1 and Flag-tagged Sp1 (**d**) or p50 (**e**). 48 hours post-transfection, cells were harvested and subjected to immunoprecipitation with an anti-Flag antibody, followed by Western blot using anti-Flag, anti-Myc, and anti-HA antibodies. The black arrows indicate the specific bands. Data in **a**–**e** are representative of at least 2 independent experiments. **f, g** 293 T cells were cotransfected with Flag-tagged Sp1 (**f**) or p50 (**g**) and Myc-tagged HDAC1 with empty vector or HA-tagged TRIM5α. Cells were treated with or without TNFα (20 ng/ml) for 24 h starting 24 h post-transfection and then harvested and subjected to coimmunoprecipitation with an anti-Flag antibody, followed by Western blot with anti-Flag, anti-Myc and anti-HA antibodies. The fold increase of interaction of Sp1-Flag/p50-Flag and HDAC1-Myc by TRIM5α over the vector control was compared between the absence or presence of TNFα treatment (**f**, **g**). **h** HDAC1-Myc and WT, ΔRING or ΔB-box 2 domain-deletion TRIM5α were cotransfected into 293 T cells. Forty-eight hours post-transfection, cells were subjected to immunoprecipitation with an anti-HA antibody, with a rabbit IgG antibody as control, followed by Western blot using anti-Myc and anti-HA antibodies (Left). The percent input of HDAC1-Myc in the immunoprecipitates was measured according to the immunoblot bands (Right). Data represents mean ± SEM (*n* = 3). The statistical significance analyses were performed using a two-sided unpaired t test. Source data are provided as a Source data file.

to detect the presence of TRIM5α at the integrated HIV-1 LTR region. Four primer pairs designed for different regions of the HIV-1 LTR (Nuc-0, DHS, Promoter, and Nuc-1) were used (Fig. 8a). Chromatin was immunoprecipitated from J-Lat cell lysates with an anti-TRIM5α antibody, with an anti-H3 antibody as a positive control and an IgG isotype antibody as a negative control. qPCR performed by all the primer pairs indicated the presence of TRIM5α in the corresponding LTR region but not in the Gag region (Fig. 8b). Additionally, coimmunoblot assays showed the interaction of TRIM5α and H3 (Fig. 8c). Treatment with

TNFα significantly reduced the occupancy of endogenous TRIM5α at the integrated HIV-1 LTR region in J-Lat cells (Fig. 8d), as well as the enrichment of ectopic TRIM5α-HA at the HIV-1 full LTR construct in transiently transfected 293 T cells (Supplementary Fig. 4), which is consistent with the results obtained from coimmunoblot assays showing that TRIM5α-promoted HDAC1–Sp1 and HDAC1–p50 interactions were decreased upon TNFα treatment (Fig. 6f, g).

Next, the enrichment of HDAC1 on the HIV-1 LTR in TRIM5α-overexpressing or TRIM5α-knockdown J-Lat cells was assessed. ChIP–

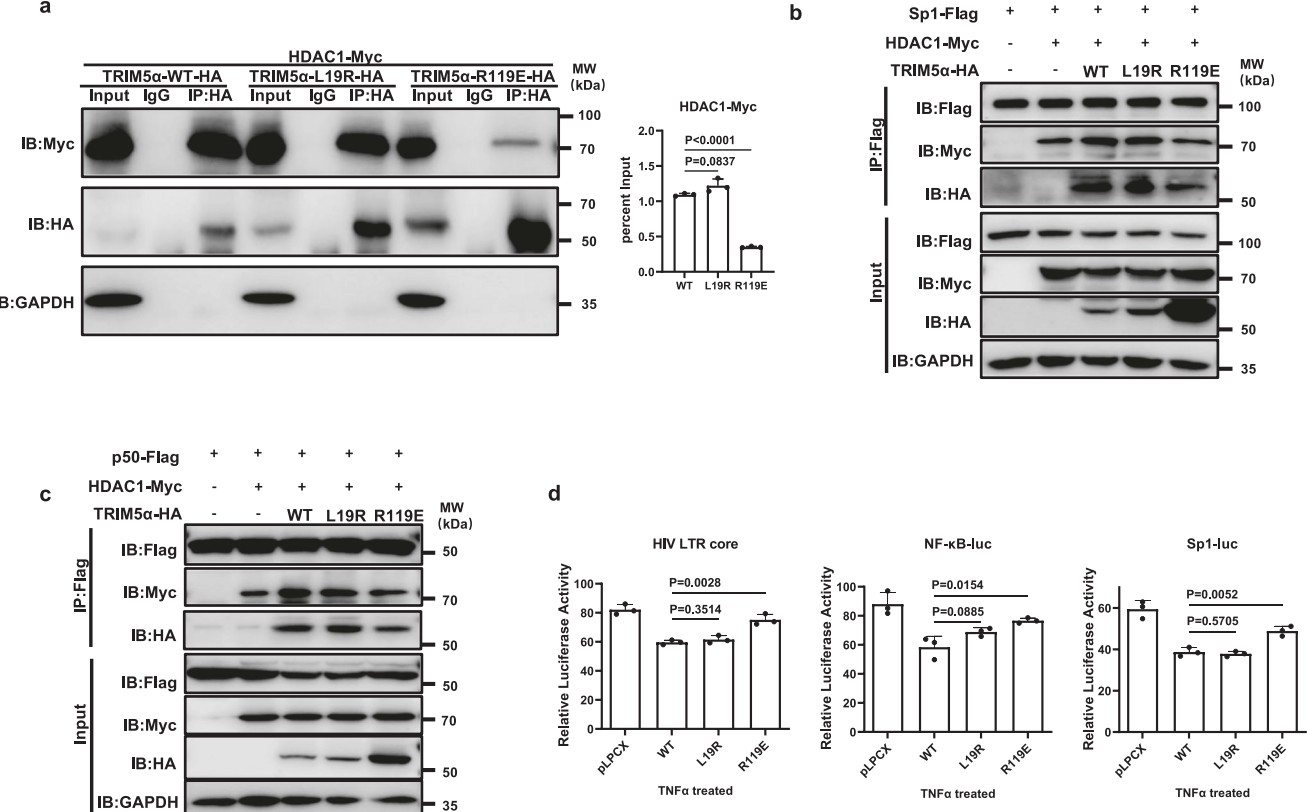

**Fig. 7 | TRIM5α R119 is the key residue mediating the interaction of TRIM5α and HDAC1, as well as promoting Sp1–HDAC1 and p50–HDAC1 interaction.**
**a** TRIM5α-WT, L19R or R119E plasmid was cotransfected with Myc-tagged HDAC1. Forty-eight hours post-transfection, cells were lysed and immunoprecipitated with a rabbit anti-HA antibody or an IgG isotype. Western blot was performed with a mouse anti-HA or a rabbit anti-Myc antibody. The percent input of HDAC1-Myc in the immunoprecipitates was measured according to the immunoblot bands (Right). **b**, **c** WT or mutant TRIM5α plasmid was cotransfected with Myc-tagged HDAC1 and Flag-tagged Sp1 (**b**) or p50 (**c**). Forty-eight hours post-transfection, cells

were harvested and subjected to immunoprecipitation with an anti-Flag antibody, followed by Western blot using anti-Flag, anti-Myc, and anti-HA antibodies. **d** Suppressive effect of TRIM5α point mutants on TNFα-driven HIV-LTR-core-luc, NF-κB-luc, and Sp1-luc reporter activities. The luciferase activity of the reporters was normalized by RL-TK. Data are representative of at least 3 independent experiments and are mean ± SD ($n = 3$). The statistical significance analyses were performed using a two-sided unpaired t test. Source data are provided as a Source data file.

qPCR analysis showed that TRIM5α-WT increased the recruitment of HDAC1 to the HIV promoter as well as to the two nucleosomes, while TRIM5α-ΔRING or TRIM5α-ΔB-box 2 had no or little such effect (Fig. 8e). On the other hand, knockdown of TRIM5α reduced the enrichment of HDAC1 on the HIV-1 LTR (Fig. 8f). We next assessed the effect of TRIM5α-mediated HDAC1 recruitment at the HIV-1 promoter on local histone acetylation levels. The levels of H3K9 acetylation at the HIV-1 Nuc-0- and Nuc-1-binding regions were decreased by the presence of TRIM5α-WT but not the domain-deletion TRIM5α mutants (Fig. 8g). Consistent with this finding, TRIM5α knockdown augmented H3K9 acetylation at the HIV-1 LTR (Fig. 8h). TRIM5α R119E, which greatly reduced the TRIM5α-mediated suppression of the LTR, showed less HDAC1 recruitment activity and a weaker impact on H3K9 acetylation than TRIM5α WT (Fig. 8i, j). We also detected the effect of TRIM5α on host NF-κB-driven promoters. ChIP–qPCR analyses using primers for IFNB1 and CXCL10 promoters showed that overexpression of TRIM5α enhanced the recruitment of HDAC1 to these promoters and reduced the levels of H3K9 acetylation (Fig. 8k), while knockdown of TRIM5α did the opposite (Fig. 8l). These results suggest that the effect of TRIM5α may be general on NF-κB-responsive genes.

Next, we determined whether TRIM5α cooperates with HDAC1 to repress HIV-1 gene transcription. Overexpression of TRIM5α potentiated the HDAC1-induced suppression of HIV-1 LTR activity, while TRIM5α-ΔRING or TRIM5α-ΔB-box 2 had little or no such effect (Supplementary Fig. 5a). However, TRIM5α depletion did not fully abolish

the effect of HDAC1 on the repression of HIV-1 full/core LTR-driven transcription (Supplementary Fig. 5b–d) or the repression of HIV-1 latency activation by HDAC1 in J-Lat cells (Supplementary Fig. 5e–f), indicating that HDAC1 inhibition of LTR transcription is partly dependent on TRIM5α.

Overall, these data support the results from the reporter assays by providing a model where the occupancy of endogenous TRIM5α at the HIV-1 promoter induces the recruitment of HDAC1 and local histone H3 deacetylation.

**Conserved suppression effects of nonhuman TRIM5α and TRIMCyp on the LTRs of exo- and endo-retroelements**
We then set out to determine whether the HIV-1 LTR suppression activity was conserved among TRIM5 orthologs, with rhesus monkey TRIM5α (rhTRIM5α) and owl monkey TRIMCyp (omTRIMCyp) as representatives. The expression of HA-tagged TRIM5 proteins was confirmed by Western blot and the results showed a consistent expression level regardless of TNFα treatment (Fig. 9a). We further examined the ectopic expression of rhTRIM5α and omTRIMCyp in the fractionated lysates from the transfected 293 T cells and found their presence in the nuclear extracts, as was seen in the huTRIM5α-expressing cells (Fig. 9b). Similar to huTRIM5α, rhTRIM5α and omTRIMCyp exhibited potent repressive effects on the TNFα-activated full-length and core LTRs of HIV-1 (Fig. 9c, d). Notably, while rhTRIM5α and omTRIMCyp activated

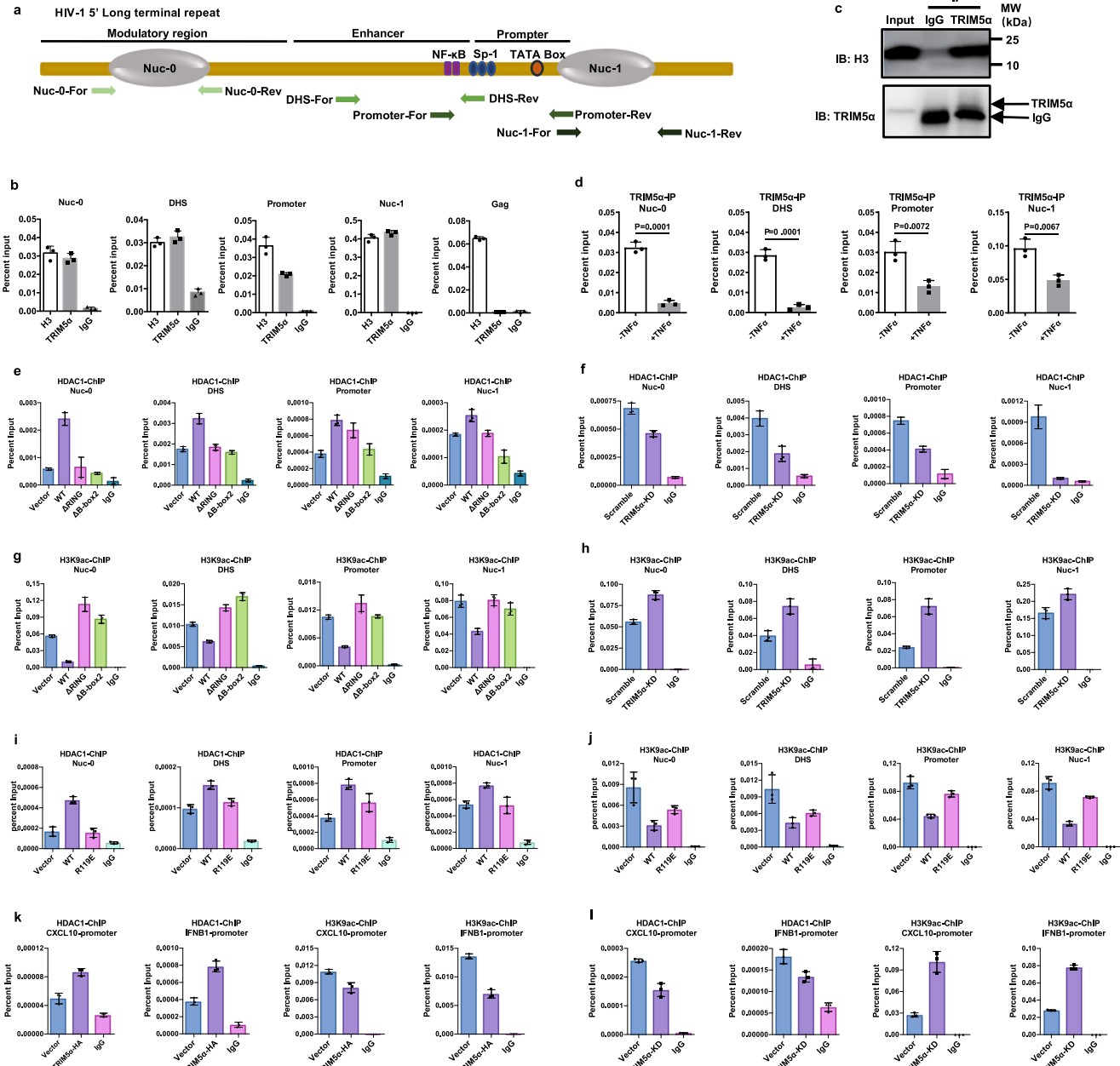

**Fig. 8 | Association of TRIM5α with HIV-1 LTR induces the recruitment of HDAC1 and local histone H3 deacetylation. a** ChIP−qPCR primers to identify different regions of the HIV-1 LTR are indicated. Nuc-0, nucleosome 0 region. Promoter, containing TATA box and the binding sites of p65/p50 and Sp1. DHS, DNase I highly sensitive region. Nuc-1, nucleosome 1 region. **b** Evidence of endogenous TRIM5α binding to HIV-1 LTR by ChIP-qPCR analysis. J-Lat cells were subjected to TRIM5α-ChIP, H3-ChIP and IgG-ChIP. Primers in **a** were used to detect the enrichment of TRIM5α on the LTR region, while primer Gag was used as a negative control. **c** Association of H3 and TRIM5α protein was detected by coimmunoprecipitation. Data are representative of at least 3 independent experiments. **d** Endogenous TRIM5α occupancy on HIV-1 LTR in the absence or presence of TNFα

(20 ng/ml) was detected by ChIP-qPCR assays. **e**, **f** J-Lat cells stably transduced with WT, ΔRING or ΔB-box 2 TRIM5α (**e**) or shRNA-TRIM5α (**f**) were stimulated by TNFα (20 ng/ml) for 30 min, and then subjected to HDAC1-ChIP and IgG-ChIP. **g**, **h** H3K9ac-ChIP and IgG-ChIP were performed in stably transduced WT or indicated domain-deletion TRIM5α (**g**) or shRNA-TRIM5α (**h**) J-Lat cells that were stimulated by TNFα (20 ng/ml) for 30 min. **i**, **j** Detection of HDAC1 enrichment (**i**) and the levels of H3K9 acetylation (**j**) at the HIV-1 LTR in the presence of WT or TRIM5α point mutants. **k**, **l** Detection of HDAC1 enrichment and the levels of H3K9 acetylation at the promoters of IFNB1 and CXCL10 in the TRIM5α-overexpressing cells (**k**) or TRIM5α-knockdown cells (**l**). The results in **b**, **d**–**l** show mean ± SD (error bars). Source data are provided as a Source data file.

the NF-κB reporter under basal conditions, which is consistent with previous studies from us and others[38,39,46], these proteins negatively regulated TNFα-mediated NF-κB activation, as seen with huTRIM5α (Fig. 9e). In contrast with the Sp1-repression activity of huTRIM5α and rhTRIM5α, omTRIMCyp exerted no negative regulatory effect on TNFα-induced Sp1-driven reporter expression (Fig. 9f), showing functional differentiation of TRIM5 orthologs in Sp1 regulation.

While HIV-1 is an exogenous retrovirus integrated into the host genome by infection, endogenous retroviruses are inherited and fixed as 8% of the human genome[47]. As reports have shown that the transcription of the HERV group is regulated by Sp1 and NF-κB[48,49], we then examined whether TRIM5α-mediated LTR repression activity held for HERV-K, the youngest and most active HERV that encodes noninfectious retroviruses[50]. The 5' LTR of HERV-K101 was used to construct the reporter construct. Again, TNFα-induced HERV-K101 LTR-

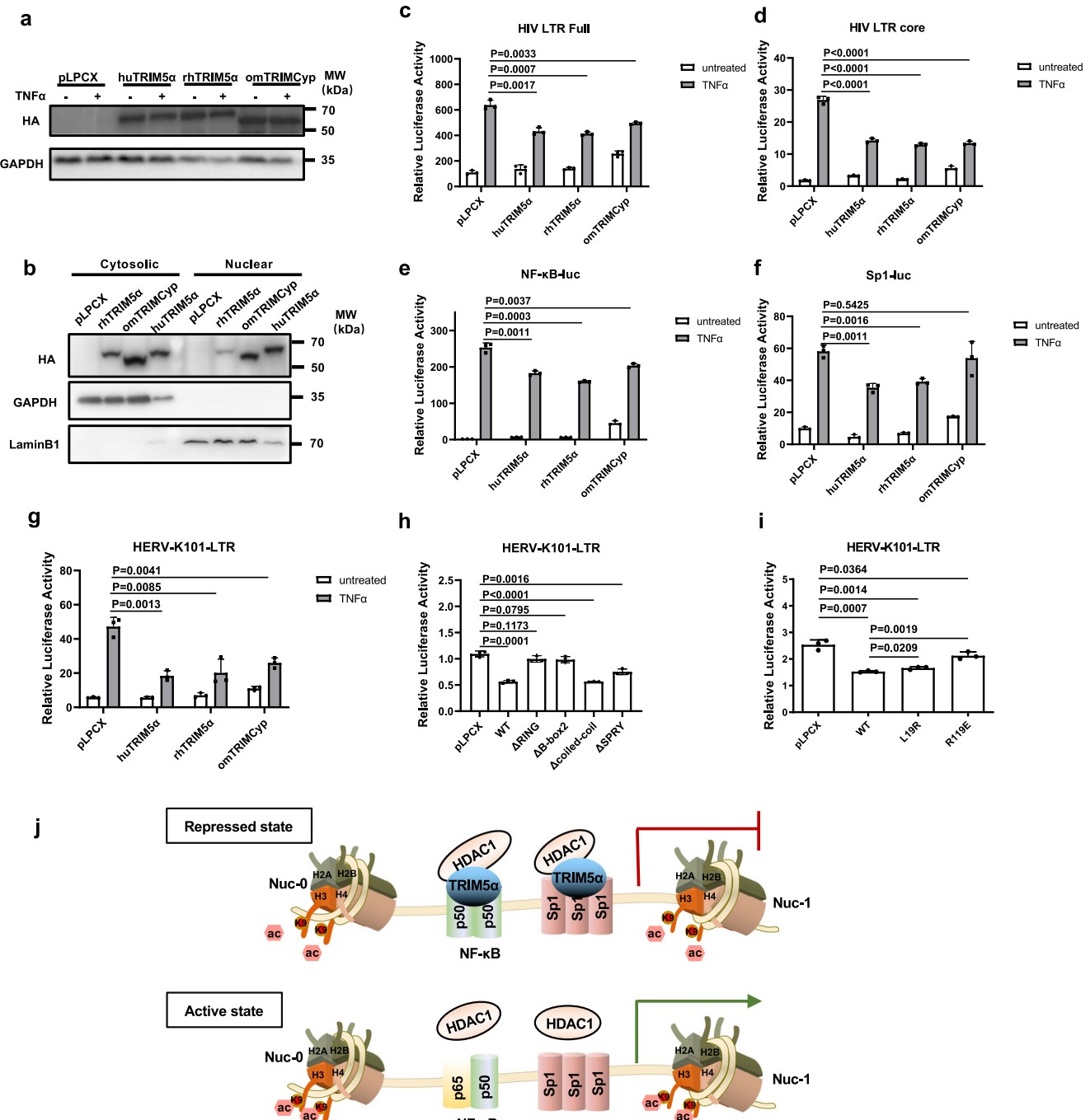

**Fig. 9 | Conserved suppression effects of TRIM5 orthologs on the LTRs of exo- and endo-retroelements. a** Exogenous expression of huTRIM5α, rhTRIM5α and omTRIMCyp in 293 T with or without the treatment of TNFα (20 ng/ml) for 24 h was shown by Western blots. **b** Cytoplasmic and nuclear extracts of 293 T cells transfected with HA-tagged rhTRIM5α, omTRIMCyp, huTRIM5α or an empty vector. Immunoblotting was performed by an anti-HA antibody, with GAPDH and LaminB1 shown as the purity of the cytoplasmic and nuclear extracts. **c–f** Empty vector or various TRIM5 proteins (100 ng) were cotransfected with pGL-HIV-LTR-full-luc (100 ng) (**c**), pGL-HIV-LTR-core-luc (100 ng) (**d**), NF-κB-luc (100 ng) (**e**), or Sp1-luc (100 ng) (**f**) along with pRL-TK. 24 h post-transfection, TNFα (20 ng/ml) was added. Luciferase activity was measured at 24 h post-stimulation. **g** The LTR reporter plasmid of HERV-K101 was cotransfected with empty vector or various TRIM5 ortholog and RL-TK in 293 T cells. Twenty-four hours post-transfection, cells

were treated with TNFα (20 ng/ml) for another 24 h, followed by luciferase activity examination. **h, i** Empty vector or various TRIM5α domain-deletion mutant (100 ng) (**h**) or TRIM5α point mutant (100 ng) (**i**) was cotransfected with HERV-K101-LTR-luc (100 ng) along with pRL-TK in 293 T. Luciferase activity was measured in cell lysates treated with TNFα (20 ng/ml) for 24 h starting 24 h post-transfection. Data are representative of at least 3 independent experiments and are mean ± SD (n = 3). The statistical significance analyses were performed using a two-sided unpaired t test. **j** Model for regulation of HIV latency by TRIM5α. At the HIV-1 latent phase, TRIM5α occupies at the HIV-1 LTR region and recruits HDAC1 to transcription factors p50 and Sp1, which leads to H3K9 deacetylation. Deletion of TRIM5α reduces HDAC1 recruitment and H3K9 deacetylation, which promotes the activation of HIV-1 gene transcription. Source data are provided as a Source data file.

driven expression was potently downregulated by all three TRIM5α orthologs (Fig. 9g). However, no suppressive effect of TRIM5α orthologs on HERV-K101 LTR was observed when TNFα stimulation was absent, with omTRIMCyp even activating basal LTR activity (Fig. 9g). By using a series of truncated forms of TRIM5α, we showed that the RING and B-box 2 domains were critical for TRIM5α-mediated repression of HERV-K101 LTR activity (Fig. 9h). In particular, alteration of R119 resulted in a significant reduction in TRIM5α's repressive activity on HERV-K101 LTR (Fig. 9i). Additionally, overexpression of TRIM5α cooperated with HDAC1 in inhibiting HERV-K101 gene transcription, while TRIM5α depletion dampened the effect of HDAC1 on the suppression of HERV-K101 LTR-driven transcription (Supplementary Fig. 5g, h). Taken together, these data suggest that TRIM5α plays a general role in safeguarding the host genome by maintaining the transcriptional silencing of exo- or endo-retroelements.

## Discussion

TRIM5α has been generally recognized as a viral capsid binding factor via the C-terminal PRY-SPRY domain, resulting in either restriction of the reverse-transcription phase or triggering of an innate immune signal to establish an antiviral state, both of which are described to function in the cytoplasm. However, several reports have shown that TRIM5α is also present in the nucleus for certain functions[28,29]. Here, we show that human TRIM5α plays an uncharacterized role by facilitating the maintenance of integrated retrovirus silencing in the nucleus. Mechanistically, TRIM5α binds to the HIV-1 LTR and promotes the recruitment of HDAC1 to the transcription factors p50 and Sp1 at the LTR region, resulting in H3K9 deacetylation (Fig. 9j and Supplementary Fig. 6).

HDAC1 is a critical factor that mediates the deacetylation of core histones and facilitates the establishment of transcriptionally repressive heterochromatin structures at the HIV LTR[4]. Several proteins have been demonstrated to recruit HDAC1 to the HIV-1 LTR, including CTIP-2, CBF-1, p50, Sp1, YY1, and FKBP3[8,9,51-53]. Among these factors, Sp1 can recruit HDAC1 via cofactors such as c-Myc or CTIP2, while whether there are cofactors promoting the recruitment of HDAC1 to p50 is unknown. Here, we report that TRIM5α interacts directly with HDAC1 and serves as a mediator that facilitates the interaction of not only HDAC1 and Sp1 but also HDAC1 and p50, providing a new mechanism for the recruitment of HDAC1 to transcription factors at the HIV-1 LTR. Although ChIP–qPCR assays have demonstrated that TRIM5α is present at the HIV-1 LTR region, whether TRIM5α is bound directly or indirectly via p50 or Sp1 is unknown. To address this, p50 or Sp1 will be deleted for the detection of TRIM5α occupancy at the HIV-1 LTR in further studies. Apart from association with p50 and Sp1, HDACs have been reported to interfere with the positive transcription elongation factor b (P-TEFb) equilibrium, as HDAC inhibitors can release P-TEFb from its inhibitory complex 7SK snRNP, leading to activation of HIV transcription[54,55]. It has been reported that JQ1 dissociates bromodomain protein Brd4 from the HIV promoter, allowing recruitment of P-TEFb to the HIV LTR[35]. Combined with our observation that the effect of TRIM5α was overwhelmed by the JQ1-mediated activation of HIV latency in J-Lat cells (Supplementary Fig. 1b), it will be of interest to ask if TRIM5α is involved in the HDAC–P-TEFb axis for HIV-1 latency establishment and maintenance.

The relationship between NF-κB and TRIM5α was initially described in studies that uncovered the innate immune sensor capacity of TRIM5α[38], wherein TRIM5α stimulates AP-1 and NF-κB signaling by catalyzing the synthesis of K63-linked ubiquitin chains that activate TAK1 in the cytosol. Consistent with this study, we observed here that TRIM5α itself (including human, rhesus macaque and owl monkey orthologs) increases basal NF-κB-driven gene expression to some extent. Notably, we uncovered in this study that TRIM5α can prevent the activation of NF-κB by promoting the recruitment of HDAC1 to the κB site at the HIV-1 LTR in a TNFα-dependent manner, suggesting that

the effect of TRIM5α on the DNA transcription level may overwhelm that on the signaling pathway. Further studies revealed that TNFα treatment released TRIM5α from the HIV-1 LTR and disrupted TRIM5α's effect on the promotion of the HDAC1–Sp1 and HDAC1–p50 interactions (Fig. 8d, Supplementary Fig. 4, and Fig. 6f, g), which may explain why the effect of TRIM5α on Sp1-luc and NF-κB-luc was exaggerated upon TNFα treatment. As many cellular promoters are driven by NF-κB, TRIM5α may participate in the transcriptional regulation of host genes in addition to viruses. ChIP–qPCR analysis using primers for IFNB1 and CXCL10 showed that TRIM5α also enhanced the recruitment of HDAC1 to these promoters (Fig. 8k, l), suggesting that the effect of TRIM5α may be general on NF-κB-responsive genes.

Expression of truncated forms of TRIM5α and analysis of the associated HDAC1 activity revealed that the interaction with HDAC1 was mediated by the N-terminal RING and B-box 2 domains of TRIM5α. The B-box 2 domain of TRIM5α has been reported to mediate self-association[18,19]; the RING domain exhibits E3 ubiquitin ligase activity and can form a dimer[56,57]. These reports indicate that both the RING and B-box 2 domains are capable of mediating protein–protein interactions. Our study revealed that the RING and B-box 2 domains are the determinants of the interactions between TRIM5α and p50, TRIM5α and Sp1, and TRIM5α and HDAC1, further promoting the formation of the p50–HDAC1 and Sp1–HDAC1 complexes and ultimately determining the fate of HDAC1-mediated HIV-1 LTR suppression. Specifically, amino acid R119, which is located in the B-box 2 domain and the previously reported key residue that mediates the higher-order assembly of TRIM5α[18,58], is mainly responsible for the interaction between TRIM5α and HDAC1, as well as TRIM5α-mediated p50–HDAC1 and Sp1–HDAC1 interactions. While HDAC1 is the major HDAC enzyme that deacetylates the histone tails at the HIV-1 LTR, HDAC2 and HDAC3 also play a role in the formation of a heterochromatin structure[51,59]. Thus, it will be of great interest to determine whether TRIM5α can bind to and compromise the recruitment of HDAC2/3 to the HIV-1 LTR.

*TRIM5* is an ancient, ubiquitously expressed, and rapidly evolving gene, with functional diversity and versatility starting to emerge. A recent study reported that TRIM5α senses and restricts long interspersed element 1, which belongs to the class of non-long terminal repeat retroelements, contributing to the integrity of the genome[60]. Our findings reveal that TRIM5α contributes not only to the transcriptional silencing of exogenous retroviruses but also to that of endogenous retroelements, supporting the concept that TRIM5α is a corepressor of viral transcription and a stabilizer of the host genome.

## Methods

### Ethics statement

The study was conducted under local ethical regulation after approval by the Medical Ethics Review Committee of Institute of Life Sciences, Chongqing Medical University. Informed consent was obtained from each healthy donor before the study.

### Reagents and antibodies

TNF-α (10602-HNAE; Sino Biological Inc.), JQ1 (SML-1524) and PMA (phorbol 12-myristate 13-acetate) (P8139; Sigma-Aldrich), Prostratin (SC-203422; Santa Cruz Biotechnology), Puromycin (ant-pr-1; Invivogen), Dynabeads Human T-Activator CD3/CD28 (CD3/28-Dynabeads) (11161D; Thermo Fisher Scientific).

Mouse anti-HA (Zen-Bioscience, 201113) (1:1000); Rabbit anti-HA (Sigma-Aldrich, H6908) (1:3000) (3μg/test); Rabbit anti-H3 (Proteintech, 17168-1) (1:3000); Rabbit anti-Myc (Proteintech, 16286-1) (1:1000) (2μg/test); Goat anti-mouse IgG-HRP (Proteintech, SA00001-1) (1:3000); Goat anti-rabbit IgG-HRP (Proteintech, SA00001-2) (1:3000); Rabbit anti-TRIM5α (Cell Signaling Technology, 14326) (1:1000) (1μg/test); Rabbit anti-HDAC1 (Cell Signaling Technology, 34589) (1:1000) (1μg/test); Rabbit anti-H3K9ac (Cell Signaling Technology, 9649) (1μg/test); Rabbit IgG isotype control (Cell Signaling

Technology, 2729) (1µg/test); Rabbit anti-HA (Cell Signaling Technology, 3724) (1:1000) (1µg/test); Mouse anti-IκBα (Cell Signaling Technology, 4814) (1:1000); Mouse anti-phospho-IκBα (Cell Signaling Technology, 2859) (1:1000); Rabbit anti-NF-κB p65 (Cell Signaling Technology, 8242) (1:1000); Rabbit anti-Sp1 (Cell Signaling Technology, 9389) (1:1000); Mouse anti-Flag (Cell Signaling Technology, 14793) (1:1000); Rabbit anti-LaminB1 antibody (Beyotime Biotec, AF5222) (1:1000); Anti-mouse Alexa Fluor 488 (Thermo Fisher Scientific, A21202) (1:2000); Anti-rabbit Alexa Fluor 555 (Thermo Fisher Scientific, A31572) (1:2000); Mouse IgG1 kappa isotype control (Thermo Fisher Scientific, 14-4714-85) (1:2000).

## Cell culture

Human embryonic kidney 293T cells (ATCC) and TZM-bl cells (NIH AIDS Reagent Program) were maintained in Dulbecco's modified Eagle's medium supplemented with 10% certified fetal bovine serum, FBS (VivaCell, Shanghai, China). HIV-1 latently infected Jurkat T cells J-Lat A10.6 (J-Lat, NIH AIDS Reagent Program) were grown in Roswell Park Memorial Institute-1640 (RPMI-1640) with 10% FBS. All cell lines were tested mycoplasma-free periodically by a PCR-based mycoplasma detection kit (Beyotime). Healthy human donors' peripheral blood mononuclear cells (PBMCs) were isolated from the buffy coat as previously described. Naive CD4 + T cells were isolated from PBMCs by MACS microbead-negative sorting and the naive CD4 + T-cell isolation kit (Miltenyi Biotec). Primary CD4 + T-cells were maintained in RPMI-1640 with 10% FBS containing 30 U/mL Recombinant human interleukin-2 (IL-2, Sino Biological Inc.).

## Plasmid construction and virus production

The pLPCX vectors encoding hemagglutinin (HA)-tagged human TRIM5α, rhesus monkey TRIM5α and owl monkey TRIMCyp were obtained from NIH AIDS reagent program. Domain-deletion mutants of human TRIM5α were generated by overlapping PCR. HIV-1 Tat was amplified from pNL4-3-luc+env-rev- and then cloned into pcDNA3.1 expression vector. pGL4.32-NF-κB-luc and pRL-TK (*Renilla* luciferase) reporter plasmids were purchased from Promega. pGL4.32-LTR-core-luc and pGL4.32-LTR-full-luc were constructed by PCR amplification of the corresponding DNA fragment from pNL4-3-luc+env-rev- and subsequently cloned into pGL4.32-NF-κB-luc by replacing NF-κB responsive element surrounded by NheI and BglII sites. ΔNF-κB-core_LTR, ΔSP1-core_LTR, and HERV-K101-LTR sequences were synthesized and then cloned into pGL4.32 vector by NheI and BglII. pGL4.32-Sp1-luc was constructed by inserting a synthesized 3* Sp1 responsive sites into pGL4.32 by NheI and BglII.

TRIM5α sequence was subcloned into the pCDH-EF1-MCS-T2A-puro vector from pLPCX-TRIM5α-HA to make pCDH-TRIM5α-HA. pCDH-ntTRIM5α-HA was generated by the point-mutation method. shRNA target sequences cloned into pLKO.1 are as follow: scramble: 5'-CCTAAGGTTAAGTCGCCCTCG-3'. TRIM5α: 5'-GCTTCCAACCTGATGCAATGT-3'. TRIM28: 5'-GCAACAGTGCTTCTCCAAAGA-3'.

Domain-deletion and point mutant constructs of TRIM5α were generated by primers as follow: huTRIM5-deltaRING-For: 5'-cgGAATTCGCCACCATG GCTTCTGGAATCCTGGTTAATGTAAAGGAGGAGGTG ATCAGTTACCAGCCTG-3', huTRIM5-Rev: 5'-GGCCTTTTATTTTATCG ATGGCTC-3'; huTRIM5-delta B-box-For: 5'-GTTGAGCCCAGAGGGCA GACAGAGGAGGTTGCCC-3', huTRIM5-delta B-box-Rev: 5'-CCCGGGCA ACCTCCTCTGTCTGCCCCTCTGGGCTC-3'; delta-coiledcoil-huTRIM 5-For: 5'-GTGGTCACCACACGTTCATCTCAGATCTGGAGC-3', delta-coiledcoil-huTRIM5-Rev: 5'-GCTCCAGATCTGAGATGAACGTGTGGTGAC CAC-3'. huTRIM5-deltaB30.2-ClaI-Rev: 5'- CCATCGATGGCTCAAGCG TAGTCTGGGACGTCGTATGGGTAGCCGCCCACTTCTAGCATTCCTTT C-3'; TRIM5α-L19R-EcoRI-For: 5'-CGgaattcGCCACCATGGCTTCTGGAA TCCTGGTTAATGTAAAGGAGGAGGTGACCTGCCCCATCTGCCGGGAA CTCCTGACA-3'; TRIM5α-R119E-For: 5'-TGGCTTTGTGAGGAGTCTCAG

GAGCACCGTGGTCAC-3', TRIM5α-R119-Rev: 5'-GTGCTCCTGAGACTCC TCACAAAGCCAGCAAATG-3'.

VSV-G-pseudotyped viruses encoding GFP were produced by triple transfection of 293 T cells using Lipofectamine 2000 (Invitrogen). HIV-1-GFP was produced as described previously[23]. VSVG-pNL4-3-luc was produced by co-transfecting pMD2.G and pNL4-3-luc+env-rev- at a 1:4 ratio.

## Generation of stably transduced cell lines

J-Lat cells were transduced with shRNA virus and cultured for 48 h followed by puromycin selection. Three days post-selection, the supernatant of infected cells was replaced with fresh RPMI-1640 and infected cells were cultured for another 2 days. 293T-TRIM5α knock-down cells and 293T-TRIM5α-HA cells were generated by transducing shTRIM5α lentivirus and pCDH-TRIM5α-HA lentivirus respectively. 1 µg/mL puromycin was used for selection.

## Generation of TRIM5α knockout cell lines

The TRIM5α gene editing was performed on HEK293T cells using CRISPR-cas9 gene editing technology. A guide RNA (sgRNA: CACCGTGGTAACTGATCCGGCACAC) was cloned into pLenti-CRISPRv2, which was kindly provided by Prof. Yi-Qun Kuang (Kunming Medical University). Lentiviral pseudotypes were generated by transfecting PMD2G, PSPAX2, sgRNA (PMD2G:PSPAX2:sgRNA=1:3:4) into $2 \times 10^5$ HEK293T cells seeded in a 6-well plate with polyethylenimine (Polysciences). After 6-8 h, medium was exchanged to DMEM containing 8% FBS. Forty-eight hours after transduction, 1 µg/ml puromycin was added to the medium. The TRIM5α knock-out monoclonal cell were isolated by limited dilution method. Once colonies were clearly visible in the wells of the 96-well plate, cells were detached by tryptic digestion and transferred to 24-well plates. The knockout phenotype was determined by Western blot.

## Flow cytometry

For HIV-1 latency-reversal assays, lentivector-transduced J-Lat cells were seeded in 24-well plates at a density of $5 \times 10^4$ cells/well and treated with indicated LRAs for 24 h before collected, washed with phosphate-buffered saline (PBS) and fixed by 4% paraformaldehyde for flow cytometry. The percentage of GFP-positive cells was examined on a BD FACSverse cytometer driven by FACSuite 1.0.3 (BD). Analysis of the acquired data was performed by FlowJo 7.6.1 (TreeStar).

## Transfection and reporter assays

293 T cells ($5 \times 10^4$) were seeded in 96-well plates one day before cotransfected with a viral LTR-driven luciferase construct or mutant constructs, pRL-TK, and TRIM5α WT or mutant-expressing plasmid using Lipofectamine 2000. Twenty-four hours post-transfection, 293 T cells were left untreated or treated with LRAs for 24 h before luciferase reporter assays were performed according to the protocol of dual luciferase reporter assay kit (Promega).

## Real-time quantitative PCR

Total cellular RNA was extracted using TRIzol reagent (TaKaRa) and reverse transcribed into cDNA using the RevertAid First Strand Crda Synthesis Kit (Thermo Fisher Scientific, USA). Real-time PCR was performed using the PowerUp SYBR SELECT MASTER MIX (Thermo Fisher) on the Bio-Rad CFX96 or ABI QuantStudio real-time PCR system. The primers used were: Gag, forward, 5'-GTG TGG AAA ATC TCT AGC AGT GG-3', and reverse, 5'-CGC TCT CGC ACC CAT CTC-3'; Tat-Rev, forward, 5'-ATG GCA GGA AGA AGC GGA G-3', and reverse, 5'-ATT CCT TCG GGC CTG TCG-3'; and GAPDH, forward, 5'-ATC CCA TCA CCA TCT TCC AGG-3'; and reverse, 5'-CCT TCT CCA TGG TGG TGA AGA C-3'. Amplifications were run as follows: initial activation was at 95 °C for 2 min, and the subsequent 40 cycles in two phases consisted of 95 °C

for 15 s and 60 °C for 30 s. The expression level was normalized with GAPDH.

### Western blot analysis and coimmunoprecipitation

To extract the whole-cell lysates, cells were lysed with RIPA lysis buffer (20 mM Tris (pH7.5), 150 mM NaCl, 1% Triton X-100) with 1 mM phenylmethylsulfonyl fluoride (Beyotime) and protease inhibitor cocktail (Thermo Fisher Scientific, USA). For nuclear and cytoplasmic fractionation, J-Lat cells with or without TNFα inductions or 293 T cells transfected with the indicated plasmids for 36 h were processed using Thermo Scientific NE-PER Nuclear and Cytoplasmic Extraction Reagents (catalog no. 78833; Thermo Fisher Scientific, USA) according to the manufacturer's protocol.

Cell lysates were separated by sodium dodecyl sulfate-polyacrylamide gel electrophoresis (SDS-PAGE) and then transferred to polyvinylidene fluoride membranes (Millipore). The blots were probed with an anti-HA (1:2500; Sigma) primary antibody, followed by an IgG-peroxidase-conjugated secondary antibody. After, blots were developed using chemiluminescent detection reagents (Millipore).

For coimmunoprecipitation analysis, cells in 10-cm culture plates were lysed in 1 ml RIPA lysis buffer with 1 mM phenylmethylsulfonyl fluoride and protease inhibitor cocktail. Cell lysates were centrifuged at $12,000 \times g$ for 10 min, and 900 μl of the clarified supernatant was used for immunoprecipitation; the remaining 100 μl lysate was diluted in 5× SDS-PAGE sample buffer, incubated at 100 °C for 5 min, and stored at −80 °C. Rabbit anti-HA or control IgG (2 μg; Sigma) was conjugated to 30 μl of Protein G-agarose beads (Santa Cruz Biotechnology) and incubated for 1 h at room temperature. The beads were centrifuged for 1 min at $12,000 \times g$, washed three times with 1 ml lysis buffer by spinning, added to 900 μl cell extract, and then incubated overnight at 4 °C. Beads were washed four times with 1 ml lysis buffer and the pellet was resuspended in 25 μl 2× SDS-PAGE sample buffer. Grayscale values were measured using ImageJ software.

### Immunofluorescence and confocal microscopy

Cells were washed with PBS and fixed in 4% paraformaldehyde for 20 min. Fixed cells were washed three times with PBS and then permeabilized with 0.1% Triton X-100 for 10 min. After washing with PBS three times, cells were blocked for 30 min at 37 °C and then incubated with a rabbit monoclonal anti-HA antibody (1:800; Cell Signaling Technology) overnight at 4 °C. Secondary antibody anti-rabbit Alexa Fluor 555 (1:1000) was applied for 2 h at room temperature. Confocal immunofluorescence microscopy was carried out using an Andor2000 microscope (Oxford instrument, UK), with a ×60 objective lens. Multiple Z-stack images were acquired using identical acquisition parameters. Obtained images were processed and analyzed with Fiji software.

### ChIP–qPCR

The ChIP–qPCR procedure was performed according to the manufacturer's instruction of SimpleChIP® Enzymatic Chromatin IP Kit (Magnet beads) (9003, Cell Signaling Technology). Briefly, $4 \times 10^6$ cells for each immunoprecipitation were cross-linked by 1% formaldehyde (12606, Cell Signaling Technology) treatment for 10 min at room temperature. The reaction was quenched by adding 2 ml of 10× glycine for 5 min. Cell pellets were washed with PBS and lysed in 1 ml ice-cold Buffer A supplemented with DTT and Protease inhibitor cocktail (PIC) and incubated on ice for 10 min. The nuclei were enriched by centrifugation at 3000 rpm for 5 min at 4 °C and resuspended in ice-cold Buffer B supplemented with DTT. Nuclei pellets were isolated by centrifuge and then digested by 0.5 μl micrococcal nuclease per IP preparation in 1 ml Buffer B supplemented with DTT for 20 min at 37 °C. The digestion was stopped by adding 50 mM EDTA followed by centrifugation at 13,000 rpm for 1 min at 4 °C. Nuclei pellet was resuspended in 1 ml ChIP Buffer

supplemented with PIC per IP preparation and incubated on ice for 10 min. The nuclei pellet was further sonicated by 3 sets of 20-s pulses using a Xinzhi sonicator (Shanghai) with a 1/8-inch probe to recover a fragmented nucleosome.

2% input sample was transferred to a new tube and stored at 20 °C before use. For immunoprecipitation of the nucleosome, 2 μl of anti-TRIM5α, anti-HDAC1, anti-H3K9ac, and anti-H3 antibodies and isotype control IgG was separately added and rotated overnight at 4 °C. Dynabeads Protein G were then added and rotated for another 2 h at 4 °C. The protein G magnetic beads were pelleted by placing the IP tubes in a magnetic separation rack and washed with 3 times low-salt washes and one-time high-salt wash. Each wash was conducted at 4 °C for 5 min under rotation. DNA enriched by protein G magnetic beads was eluted by ChIP Elution Buffer for 30 min at 65 °C with gentle vortexing. All the DNA samples including 2% input samples were reversed cross-linking with 200 mM NaCl and Proteinase K at 65 °C for 2 h. DNA was then purified by spin columns (14209, Cell Signaling Technology).

For qPCR of immunoprecipitated DNA, targeting primers were used to analyze the protein-binding and histone modification at HIV-1 genes. The oligonucleotide primers for ChIP–qPCR were: HIV-1 Promoter (−116) forward, 5′-AGC TTG CTA CAA GGG ACT TTC C, and (+4) reverse, 5′-ACC CAG TAC AGG CAA AAA GCA G; HIV-1 Nuc 0 forward, 5′-TGG ATC TAC CAC ACA CAA GG-3′ and reverse, 5′-GTA CTA ACT TGA AGC ACC ATC C-3′; HIV-1 DHS forward, 5′-AAG TTT GAC AGC CTC CTA GC-3′ and reverse, 5′-CAC ACC TCC CTG GAA AGT C-3′; HIV-1 Nuc 1 forward, 5′-GCTTTTTGCCTGTACTGGGTCTCTC-3′ and reverse, 5′-CC ACACTGACTAAAAGGGTCTGAGG-3′; HIV Gag forward, 5′-GCA GAG AG G CAA TTT TAG GAA CC-3′ and reverse, 5′-CCT TTT TCC TAG GGG CCC TG-3′; CXCL10 forward: 5′-GGGTATCTGATTTGTGGTG-3′ and reverse, 5′-AGGTCTGTCTCTATGCGTG-3′; IFNB1 forward: 5′-CCCACTTTCAC TTCTCCC-3′ and reverse, 5′-AAAACATTAGAAAACCTCACAG-3′.

### HIV-1 latently infected primary CD4 + T cell model

HIV-1 latent infection was done following a previous report[61]. Briefly, PBMCs were isolated using Ficoll density gradient centrifugation (GE Life) from buffy coats of two healthy anonymous donors. Naïve CD4 + T cells were isolated by a naïve human CD4 + T-cell enrichment kit (Miltenyi Biotec). CD4 + T cells were cultured with CD3/28-Dynabeads for 3 days; the Dynabeads were then removed and cultured with IL-2 (30 IU/ml) for 4 days. CD4 + T cells were infected with NL4-3/Luc pseudovirus and were then washed with prewarmed RPMI with IL-2 and plated on a 12-well plate the next day. At 7 dpi, cells were transduced with shRNA-scramble or shRNA-TRIM5α lentiviruses, followed by 500 ng/ml puromycin treatment at 10 dpi. A portion of the cells was used for Western blotting to check the knock-down efficiency of shTRIM5α at 13 dpi before cells were reactivated using CD3/28-Dynabeads. Three days after reactivation, cells were washed with PBS and lysed by 100 μl passive lysis buffer for luciferase assay. The luciferase activity in cell extracts was quantified by mixing 50 μl of lysate with 50 μl of substrate (Luciferase Assay System, Beyotime).

### Statistical analysis

Data were analyzed as mean ± SD unless otherwise stated. The statistical significance analyses were performed using two-sided unpaired $t$ test ($P$ values). Analyses were performed by using Prism 6 software (GraphPad).

### Reporting summary

Further information on research design is available in the Nature Portfolio Reporting Summary linked to this article.

## Data availability

Source data are provided with this paper.

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

## Acknowledgements

We thank Dr. Ping Lu (University of Massachusetts Medical School) for critical reading and constructive suggestions of this manuscript. This work was supported by grants from the Scientific and Technological Research Program of Chongqing Municipal Education Commission (KJQN202000424), Chongqing Natural Science Foundation (cstc2021jcyj-msxmX0253), Project of Yunnan Applied Basic Research Project-Kunming Medical University Union Foundation (201901D070045), the National Natural Science Foundation of China (82160388), and the program Innovative Research Team in Science and Technology in Kunming Medical University (CXTD202202).

## Author contributions

D.M. conceived, designed and performed experiments, analyzed the data, and wrote the manuscript. X.-H.R., J.-W.Z., R.-Z.N., Y.-Y.C., and W.-H.Z. performed experiments and analyzed the data. J.-W.Z. revised the manuscript. Y.-T.Z. contributed to plasmids and cells. All authors read and approved the final manuscript.

## Competing interests

The authors declare no competing interests.
