## [Peer review file · Nature Communications]

TRIM5 α recruits HDAC1 to p50 and Sp1 and promotes H3K9 deacetylation at the HIV-1 LTRREVIEWER COMMENTS

Reviewer #1 (Remarks to the Author):

TRIM5 has previously been shown to have restrictive effects on retroviral replication. TRIM5a is known to affect NF- κ B signaling. This study convincingly shows that TRIM5a can recruit HDAC1 to the activated HIV-1 promoter, identifying a novel mechanism of establishing proviral latency. The authors have specifically identified the regions that mediate this protein-protein interaction. Overall this is a very interesting and potentially important manuscript. This study is well-conducted and conclusions are largely supported by the data. Some points that could strengthen the manuscript are presented below. The work is of high significance to the field as it points to a way in which HIV latency is established, even when the provirus is integrated in permissive chromatin regions.

Major points:

1. It is important for the conclusions of the manuscript that TRIM5a is present in the nucleus. Previous reports have shown TRIM5a being mainly cytoplasmic. References are provided, but they do not support sustained high levels of TRIM5a in lymphocyte-derived cells. It is unexpected that the authors show the same TRIM5a protein levels in the cytoplasm and nucleus in J-lat cells in Fig 4g. Given the importance of the protein localization, stronger evidence for TRIM5a in the nucleus is required.

A possible explanation could be that the shRNA introduction in itself is responsible for TRIM5a nuclear localization. The shRNAs are introduced through lentiviral transduction, and TRIM5a could then be engaged by that, affecting both scramble and T5-KD. Including the parental (untransduced) J-lat cells here would strengthen the conclusion.

2. Much of the data is generated in 293T cells, cells that are not targeted by HIV-1. However, in Fig 2f, the authors present important data on primary cells. In this primary cell model, the authors show that knock down of TRIM5a leads to increase of reporter luc-expression.

When cells are grown ex vivo for 16 days, cell viability is usually rather low. An effect on viability of TRIM5a knockdown would in turn affect HIV activation. To remove this concern, the authors could report the cell viability in the different conditions. The level of luciferase activity in unstimulated cells should also be presented. Importantly, this data is from a single individual donor. To draw general conclusions, cells from a few more individuals should be included.

3. Is the role of TRIM5a exclusive to HIV-1 and viral elements or is this a general effect also active on host genes? The authors touch upon this issue (line 388-390) but never explore it. In the abstract, the identified mechanism is suggested as "a target for future HIV eradication strategies". This would require the mechanism to predominantly act on non-host elements. In the current study, HIV-1 could easily be seen as a reporter gene regulated by NF κ B and SP1. Are also host loci under NF κ B control, such as e.g. TNF α , affected by the TRIM5a-HDAC1 interaction? This could be tested by adding a few more primers in the ChIP-PCR experiments (selected parts of Fig 7).

Minor points:

1. Which J-lat clone has been used?

2. It is suggested that "TRIM5a suppresses HIV-1 latency reactivation" (line 112). Instead of presenting a TRIM5a inhibitor as a new LRA, the implications of this mechanism seem to be broader. Given the proposed HDAC-mediated mechanism, TRIM5a recruits HDAC1 to the already active HIV-1 promoter, and subsequently silences it. Apart from TRIM5a having a role in the initial establishment of proviral latency when the provirus is integrated in a chromatin region of acetylated histones, it would also re-silence activatable proviruses which gain acetyl marks during activation. These toggling proviruses poses a main hinder for an HIV-1 cure.

3. The amount of experiments presented makes the text rather dense. Focusing on the more important aspects in the main figures and putting more peripheral observations in the supplementary material would make the study more accessible.

Reviewer #2 (Remarks to the Author):

The authors show that depletion of TRIM5 in human cells increases basal and TNF-induced transcription from an integrated HIV-1 LTR in T cell lines including Jurkat CD4+ cells, and HeLa cells. The effect was reverse by restoration of TRIM5 expression with a transgene, supporting a TRIM5-specific effect. The effect was mapped to the NF- κ B and Sp1 sites in the HIV-1 LTR, but not interference with I κ B α phosphorylation, and to the TRIM5 RING and B-Box domains. Mechanistically, TRIM5 promoted recruitment of HDAC1 to Sp1 and NF- κ B proteins, and HDAC1 to the LTR, which was translated in a reduction in local H3K9 acetylation. TRIM5 itself was bound to HIV LTR-containing DNA. Repression of HERV-K was also observed.

Figure 7 contains perhaps some of the more compelling data in this paper, showing TRIM5 association with viral DNA, which is a surprising finding not yet predicted by the current TRIM5 literature. However, this reviewer has several concerns regarding the presentation and interpretation of the data in the manuscript, as outlined below. Overall, some of the suppressive phenotypes are inconsistent, and the conclusions not always supported by the data as shown, in some cases misinterpreted. The Western blots are often difficult to interpret and therefore do not always support the conclusions reached.

An effect of TRIM5 on expression of integrated HIV provirus is surprising given its potent capsid binding activity and restriction of early steps of retrovirus infection. TRIM5 has not previously been implicated in this stage of the HIV lifecycle. If true, this would represent a significant advance for the field, however further data is required to support some of the claims made. Is there any evidence that TRIM5 expression correlates with HIV quiescence in vivo?

Major comments:

General:

Western blots:

- Molecular weight indicators should be displayed on all western blot panels for it to be possible to interpret them.
- Exposure – e.g. fig 4h tubulin, it's impossible to interpret as the exposure is too high. This is a recurrent concern.

How were TRIM5 mutants constructed – were these deletions, what were the domain boundaries? How was proper protein folding accounted for or is for example the Δ Box simply misfolded? If interference with higher order assembly or E3 ligase is required, single point mutants informed by domain structures are now known, for example B-Box R119E in human TRIM5 completely abolishes higher order assembly but not inherent E3 of the RING domain. In the RING domain, E11R or L19R selectively disable E2 binding and thus RING activity, without impacting on overall restriction of retroviral infection. Other mutants are available. Such mutants should be employed to narrow the effect on specific TRIM5 function, otherwise these deletion mutants might simply be non-functional generally and thus not informative.

Figure 1

1d – pg6 line 112 – where is the data showing dose-dependent repression?

P6 line 120 – Query the interpretation of 1d. Jq1 is the only LRA that loses its phenotype in TRIM5 depleted cells, suggesting that TRIM5 repression and the target of Jq1 are mechanistically linked? Indeed, one model could be that TRIM5 suppresses transcription elongation, rendering JQ1 ineffective in the KD cell. The conclusion in the text 'suggesting that TRIM5 had no effect on transcription elongation' is not supported by the data.

Figure 2

Pg 8 Line 155 – 7c – the interpretation is questioned. The data to this reviewer suggests that LRAs (Latency reversing agents) and TRIM5 work independently, as the drugs work as effectively in WT

or KD cells, but the base line is higher in the KD scenario.

Figure 3

Switched GFP for Luc – not clear why – so essentially similar data as presented in Figures 1-2.

What is the basis for switching between a stable LTR suppression assay as in Figures 1-2, and a transient LTR expression as employed here. In Figure 3, cotransfection of TRIM5 and LTR-HIV constructs mean that the regulation at the LTR is likely to be wholly distinct to the regulation measured in the chronically infected scenario. Some comment on this would be valuable.

3c – Interesting result. How is TRIM5 only able to antagonise TNF-induced transcription and not Tat driven elongation, when the latter requires transcription initiation and production of the TAR element to which Tat binds (in complex with CDK9 and Cyclin T1)? What is the logic for this experiment?

3d – we can't see the values in untreated conditions. Log scale would be clearer. How were these graphs normalised (if at all)?

3e – empty plasmid control is missing. How were these values normalised to one another?

3c-3e – why are the RLU units so different between these experiments?

In 3h-i, there seems to be a TNF-independent TRIM5 phenotype. In 3m-n, there seems to be a TNF-dependent TRIM5 phenotype. Would the authors comment on this? Both assays involved TRIM5 depletion.

The authors do not entertain the possibility of clonal artefact. Either multiple KO clones should be compared, or 1-2 clones should be reconstituted with TRIM5 to restore the parental phenotype. Otherwise, clone 10# could just be moderately more sensitive to TNF than the parental line.

Figure 4

Same concern as above re: clonal effect.

Page 11, line 201, TRIM5 activates the Δ SP1 core LTR presumably because TRIM5 stimulates NFkB expression when transfected into 293T cells.

4c – we can't see the black bars or any change between conditions, either separate panels or a log panel should be used, although this might alter the conclusions made?

4g – Pg 12 line 224 – it's not possible to draw this conclusion from the data presented, the p65 blot is too dirty to see the NFkB band in the nuclear fractions.

4h – p-IkB α blot too dirty to draw conclusion.

Figure 5

It should be noted that TRIM5 itself stimulates an NFkB signal in 293T cells. The phenotype of the mutants used is inversely correlated with their NFkB activity (via TRIM5 E3 activity). Is this a coincidence or relevant to the phenotype?

Figure 6

Generally the blots are overexposed making their interpretation difficult.

For the majority of panels in this figure there is a lack of an IgG control antibody that is important to know that the IPs are specific. Otherwise, the mirroring of input with IP could simply be non-specific binding of proteins to the protein A agarose etc.

6f – pg14 line 274 – in fact, it looks like the effect of TRIM5 on HDAC1-sp1 binding is weaker in the presence of TNF, and not equivalent as suggested in the text.

Figure 7

7j-m – this data appear to be misinterpreted. TRIM5 KO does not rescue the inhibition caused by HDAC1 expression on LTR transcription. Rather, the fold inhibition caused by HDAC1 is equivalent in control or TRIM5 KD cells. In 7 m there seems to be negligible effect caused by HDAC1.

Minor comments:

Pg 7 line 129 and line 137 – 1g, not 1e?

Page 4 line 60 – the sentence is a little ambiguous. The TRIM5 RING is widely known to be an adaptor type E3 for ubiquitination, but not Sumoylation or ISGylation (as in, biochemical evidence, not just overexpressed proteins in cells). The RING fold in general however is involved in all 3 of these. So just to clarify whether this sentence relates to the RING fold in general, or specifically the TRIM5 RING.

'PRY-SPRY' not just 'SPRY' – e.g. Page 3 line 59

Point-by-Point Response to Reviewers' Comments

Manuscript number: NCOMMS-22-43642

Title: TRIM5 α recruits HDAC1 to p50 and Sp1 and promotes H3K9 deacetylation at the HIV-1 LTR

Reviewer #1 (Remarks to the Author):

TRIM5 has previously been shown to have restrictive effects on retroviral replication. TRIM5 α is known to affect NF- κ B signaling. This study convincingly shows that TRIM5 α can recruit HDAC1 to the activated HIV-1 promoter, identifying a novel mechanism of establishing proviral latency. The authors have specifically identified the regions that mediate this protein-protein interaction.

Overall this is a very interesting and potentially important manuscript. This study is well-conducted and conclusions are largely supported by the data. Some points that could strengthen the manuscript are presented below. The work is of high significance to the field as it points to a way in which HIV latency is established, even when the provirus is integrated in permissive chromatin regions.

Response: Thank you very much for your time spent reviewing this manuscript. We appreciate your positive comments and valuable suggestions that help to improve the quality of our manuscript. We have conducted new experiments and made modifications according to the suggestions. We hope that the revised manuscript has fully addressed your concern.

Major points:

1. It is important for the conclusions of the manuscript that TRIM5 α is present in the nucleus. Previous reports have shown TRIM5 α being mainly cytoplasmic. References are provided, but they do not support sustained high levels of TRIM5 α in

lymphocyte-derived cells. It is unexpected that the authors show the same TRIM5 α protein levels in the cytoplasm and nucleus in J-lat cells in Fig 4g. Given the importance of the protein localization, stronger evidence for TRIM5 α in the nucleus is required.

A possible explanation could be that the shRNA introduction in itself is responsible for TRIM5 α nuclear localization. The shRNAs are introduced through lentiviral transduction, and TRIM5 α could then be engaged by that, affecting both scramble and T5-KD. Including the parental (untransduced) J-lat cells here would strengthen the conclusion.

Response: We thank the reviewer for the helpful comment. As the reviewer suggested, we examined the expression of TRIM5 α in the cytoplasmic and nuclear extracts from parental (untransduced) J-Lat cells. The results showed that TRIM5 α existed not only in the cytoplasm but also in the nuclear extract of the parental J-Lat cells. Furthermore, we used primary human PBMCs and 293T cells to detect TRIM5 α protein distribution. Again, TRIM5 α was found in the cytoplasm and nucleus of both PBMCs and 293T cells. Notably, the nuclear extract of PBMCs harbored substantial amounts of TRIM5 α , suggesting that TRIM5 α may play an important role in PBMCs. We have added these results in Revised Figure 4h. We also added a description in Lines 232-239, Page 12.

Revised Figure 4h

2. Much of the data is generated in 293T cells, cells that are not targeted by HIV-1. However, in Fig 2f, the authors present important data on primary cells. In this primary cell model, the authors show that knock down of TRIM5 α leads to

increase of reporter luc-expression.

When cells are grown *ex vivo* for 16 days, cell viability is usually rather low. An effect on viability of TRIM5 α knockdown would in turn affect HIV activation. To remove this concern, the authors could report the cell viability in the different conditions. The level of luciferase activity in unstimulated cells should also be presented. Importantly, this data is from a single individual donor. To draw general conclusions, cells from a few more individuals should be included.

Response: We thank the reviewer for the helpful comment. In our new experiment with primary CD4⁺ cells from two more donors, we compared the cell viability in the scramble condition and TRIM5 α knockdown condition by CCK8 assays. The results showed that the cell viability was comparable between the scramble and the TRIM5 α knockdown cells (shown below in Rebuttal Figure 1), suggesting that TRIM5 α KD had little effect on cell viability compared to scramble-KD, which may not in turn affect HIV activation.

Rebuttal Figure 1

The level of luciferase activity in unstimulated cells (0 day) is now presented in Revised Figure 3d.

Revised Figure 3d

As the reviewer suggested, we conducted *in vivo* assays using CD4⁺ cells from two more donors. The new results are included in Revised Figure 3d-f.

Revised Figure 3d-f

3. Is the role of TRIM5α exclusive to HIV-1 and viral elements or is this a general effect also active on host genes? The authors touch upon this issue (line 388-390) but never explore it. In the abstract, the identified mechanism is suggested as “a target for future HIV eradication strategies”. This would require the mechanism to predominantly act on non-host elements. In the current study, HIV-1 could easily be seen as a reporter gene regulated by NFκB and SP1. Are also host loci under NFκB control, such as e.g. TNFα, affected by the TRIM5α-HDAC1 interaction? This could be tested by adding a few more primers in the ChIP-PCR experiments (selected parts of Fig 7).

Response: We thank the reviewer for the constructive suggestion. We performed ChIP-qPCR experiments by using primers for IFNB1 and CXCL10, which are two representative genes under NFκB control. The results showed that TRIM5α-WT but not TRIM5α-ΔRING or TRIM5α-ΔB-box 2 enhanced the recruitment of HDAC1 to the promoters of IFNB1 and CXCL10. Consistently, knockdown of TRIM5α reduced the enrichment of HDAC1 on the promoters of IFNB1 and CXCL10.

We also detected the effect of TRIM5α on the levels of H3K9 acetylation at the

promoters of IFNB1 and CXCL10. The results showed that the local histone acetylation levels of these promoters were compromised by overexpression of wild-type TRIM5 α but significantly elevated by knockdown of TRIM5 α . Please find these results in Revised Figure 8k, i.

Revised Figure 8k, i

Altogether, these data indicate that the effect of TRIM5 α is general on NF κ B-responsive genes. Thus, we toned down the conclusion that TRIM5 α is “a target for future HIV eradication strategies” in the abstract. We have revised the sentence from “These findings provide new insights into the molecular mechanisms by which histone deacetylases are recruited to the retroviral promoters and a target for future HIV eradication strategies” to “These findings provide new insights into the molecular mechanisms by which proviral latency is initially established and activatable proviruses are resiled by histone deacetylase recruitment”; please see Lines 16-18, Page 1. We have also added a description of the new results in Lines 436-440, Page 22.

Minor points:

1. Which J-lat clone has been used?

Response: We have used J-Lat A10.6. We have added the full name of this J-Lat

clone in the “Materials and Methods”; please see Lines 490-491, Page 26.

2. It is suggested that “TRIM5a suppresses HIV-1 latency reactivation” (line 112). Instead of presenting a TRIM5a inhibitor as a new LRA, the implications of this mechanism seem to be broader. Given the proposed HDAC-mediated mechanism, TRIM5a recruits HDAC1 to the already active HIV-1 promoter, and subsequently silences it. Apart from TRIM5a having a role in the initial establishment of proviral latency when the provirus is integrated in a chromatin region of acetylated histones, it would also re-silence activatable proviruses which gain acetyl marks during activation. These toggling proviruses poses a main hinder for an HIV-1 cure.

Response: We thank the reviewer for the helpful comment. We agree with the reviewer that TRIM5a has a role in the initial establishment of proviral latency when the provirus is integrated in a chromatin region of acetylated histones, and it also resiliences activatable proviruses that gain acetyl marks during activation. We highlight this conclusion in the abstract as follows: “These findings provide new insights into the molecular mechanisms by which proviral latency is initially established and activatable proviruses are resilienced by histone deacetylase recruitment”; please see Lines 16-18, Page 1.

3. The amount of experiments presented makes the text rather dense. Focusing on the more important aspects in the main figures and putting more peripheral observations in the supplementary material would make the study more accessible.

Response: We thank the reviewer for the helpful comment. We have now moved Figure 1d to Supplementary Figure 1, Figure 7h-m to Supplementary Figure 4, and Figure 8i-j to Supplementary Figure 4 to make the study clearer, more focused and more accessible.

Reviewer #2 (Comments for the Author):

The authors show that depletion of TRIM5 in human cells increases basal and TNF-induced transcription from an integrated HIV-1 LTR in T cell lines including Jurkat CD4+ cells, and HeLa cells. The effect was reverse by restoration of TRIM5 expression with a transgene, supporting a TRIM5-specific effect. The effect was mapped to the NF- κ B and Sp1 sites in the HIV-1 LTR, but not interference with I κ Ba phosphorylation, and to the TRIM5 RING and B-Box domains. Mechanistically, TRIM5 promoted recruitment of HDAC1 to Sp1 and NF- κ B proteins, and HDAC1 to the LTR, which was translated in a reduction in local H3K9 acetylation. TRIM5 itself was bound to HIV LTR-containing DNA. Repression of HERV-K was also observed.

Figure 7 contains perhaps some of the more compelling data in this paper, showing TRIM5 association with viral DNA, which is a surprising finding not yet predicted by the current TRIM5 literature. However, this reviewer has several concerns regarding the presentation and interpretation of the data in the manuscript, as outlined below. Overall, some of the suppressive phenotypes are inconsistent, and the conclusions not always supported by the data as shown, in some cases misinterpreted. The Western blots are often difficult to interpret and therefore do not always support the conclusions reached.

An effect of TRIM5 on expression of integrated HIV provirus is surprising given its potent capsid binding activity and restriction of early steps of retrovirus infection. TRIM5 has not previously been implicated in this stage of the HIV lifecycle. If true, this would represent a significant advance for the field, however further data is required to support some of the claims made. Is there any evidence that TRIM5 expression correlates with HIV quiescence in vivo?

Response: We feel great thanks for your time and your professional work on this manuscript. We appreciate your valuable comments that helped to improve the quality of our manuscript. We have conducted new experiments and made extensive revisions

according to the constructive suggestions. We hope that the revised manuscript has fully addressed your concern.

Regarding the question “*Is there any evidence that TRIM5 expression correlates with HIV quiescence in vivo?*”, we performed some literature research and found that there is one study published in 2021 reporting significantly higher levels of TRIM5 α expression in cells from long-term nonprogressors (LTNP) with respect to HIV-1-infected normal progressor patients¹. This study suggests that TRIM5 α may correlate with HIV transcription silencing, yet further investigation is needed.

Major comments:

General:

1. Western blots:

- *Molecular weight indicators should be displayed on all western blot panels for it to be possible to interpret them.*

Response: We thank the reviewer for the helpful comment. We have now added molecular weight indicators to all western blot panels.

- *Exposure – e.g. fig 4h tubulin, it’s impossible to interpret as the exposure is too high. This is a recurrent concern.*

Response: We thank the reviewer for the helpful comment. We have now replaced the overexposed blots with properly exposed blots for a better interpretation.

2. How were TRIM5 mutants constructed – were these deletions, what were the domain boundaries? How was proper protein folding accounted for or is for example the Δ Box simply misfolded? If interference with higher order assembly or E3 ligase is required, single point mutants informed by domain structures are now known, for example B-Box R119E in human TRIM5 completely abolishes higher order assembly

but not inherent E3 of the RING domain. In the RING domain, E11R or L19R selectively disable E2 binding and thus RING activity, without impacting on overall restriction of retroviral infection. Other mutants are available. Such mutants should be employed to narrow the effect on specific TRIM5 function, otherwise these deletion mutants might simply be non-functional generally and thus not informative.

Response: We thank the reviewer for these helpful comments. We apologize for not clearly presenting the construction strategy of the TRIM5 mutants. We have now indicated the domain boundaries in the schematic of TRIM5 mutants in Revised Figure 5a for better clarity.

Revised Figure 5a

We agree with the reviewer that domain truncation may cause misfolded and nonfunctional proteins. As the reviewer suggested, we employed single-point mutants B-Box R119E, which interferes with higher-order assembly, and L19R, which disables E2 binding and thus RING activity (we did not use E11R because the ΔRING construct, which was devoid of aa 15-58, impaired the suppression activity of TRIM5 α , suggesting that E11 does not contribute to TRIM5 α inhibition of LTR activation), to determine whether higher-order assembly or RING activity accounts for TRIM5 α -mediated LTR regulation.

The Co-IP assay results showed that R119E could indeed significantly impair the interaction of HDAC1 and TRIM5 α -WT, as well as the TRIM5 α -mediated HDAC1-Sp1 and HDAC1-p50 interactions, while L19R was not able to do so (Revise Figure 7a-c). Moreover, luciferase assays showed that R119E but not L19R greatly reduced the TRIM5 α -mediated suppression of LTR-luc, NFkB-luc, and Sp1-luc (Revise Figure 7d).

Revised Figure 7

ChIP-qPCR analysis further showed that R119E exhibited impaired ability to enhance the recruitment of HDAC1 to the HIV LTR and decreased impact on the levels of H3K9 acetylation compared to TRIM5 α -WT, as shown in Revised Figure 8i, j.

Revised Figure 8i, j

Altogether, the findings demonstrated that R119, the previously reported key residue that mediates the higher-order assembly of TRIM5 α ^{2, 3}, is also the main residue that mediates the interaction of HDAC1 and TRIM5 α . RING activity, on the other hand, is not related to the interaction between HDAC1 and TRIM5 α . The key amino acid(s) in the RING domain that promote the interaction between HDAC1 and TRIM5 α have yet to be defined by further endeavors.

Again, we deeply appreciate the reviewer for the constructive advice that helps to reveal a more detailed molecular mechanism, which improves the quality of this manuscript. We included figures of these new results in Revised Figures 7 and 8. We have also added a description of these new results in Lines 289-303, Pages 15-16, and Lines 335-343, Pages 17-18.

3. Figure 1

1d – pg6 line 112 – where is the data showing dose-dependent repression?

Response: We thank the reviewer for pointing this out. We apologize for describing 2 dose treatments as a “dose-dependent” manner, and we have revised the sentence as follows: “Furthermore, we showed that TRIM5 α suppressed HIV-1 latency reactivation in the presence of different dosages of TNF α treatment”. Please see Lines 112-113, Page 6.

P6 line 120 – Query the interpretation of 1d. Jq1 is the only LRA that loses its phenotype in TRIM5 depleted cells, suggesting that TRIM5 repression and the target of Jq1 are mechanistically linked? Indeed, one model could be that TRIM5 suppresses transcription elongation, rendering JQ1 ineffective in the KD cell. The conclusion in the text ‘suggesting that TRIM5 had no effect on transcription elongation’ is not supported by the data.

Response: We thank the reviewer for pointing this out. We agree with the reviewer that JQ1 losing its phenotype in TRIM5-depleted cells suggests that TRIM5 repression and the target of JQ1 are mechanistically linked. It was reported that the mechanism by which JQ1 activates HIV-1 latency is that JQ1 dissociates bromodomain protein Brd4 from the HIV promoter, allowing recruitment of P-TEFb to the HIV LTR⁴. There are also studies reporting that the dissociation of HDAC releases P-TEFb from its inhibitory complex, which leads to the activation of HIV transcription^{5,6}. Therefore, it is possible that JQ1 treatment may saturate the effect of

TRIM5 α if it is involved in the HDAC–P-TEFb axis. In the present study, we focused on how TRIM5 α recruits HDAC1 to p50 and Sp1 and to the HIV-1 LTR. In the near future, we will investigate whether TRIM5 α regulates HIV latency in an HDAC–P-TEFb-dependent manner.

Therefore, we have revised the text from “suggesting that TRIM5 had no effect on transcription elongation” to “suggesting that the effect of JQ1-mediated activation saturates the effect of TRIM5 α ”; please see Lines 120-121, Page 6. We also discussed this finding in the Discussion section: “Apart from association with p50 and Sp1, HDACs have been reported to interfere with the positive transcription elongation factor b (P-TEFb) equilibrium, as HDAC inhibitors can release P-TEFb from its inhibitory complex 7SK snRNP, leading to activation of HIV transcription. It has been reported that JQ1 dissociates bromodomain protein Brd4 from the HIV promoter, allowing recruitment of P-TEFb to the HIV LTR. Combined with our observation that the effect of TRIM5 α was overwhelmed by the JQ1-mediated activation of HIV latency in J-Lat cells, it will be of interest to ask if TRIM5 α is involved in the HDAC–P-TEFb axis.”; please see Lines 418-426, Pages 21-22.

4. Figure 2

Pg 8 Line 155 – 7c – the interpretation is questioned. The data to this reviewer suggests that LRAs (Latency reversing agents) and TRIM5 work independently, as the drugs work as effectively in WT or KD cells, but the base line is higher in the KD scenario.

Response: We thank the reviewer for the helpful comment. We agree with the reviewer that these data suggest that these LRAs and TRIM5 work independently to some extent. However, the drugs did not work as effectively as in WT or KD cells, as the fold change in TRIM5KD-mediated latency activation in all LRA-treated cells was less than that in the untreated control (shown below in Rebuttal Figure 2). Hence, we conclude that the targets of these LRAs overlap with the targets of TRIM5.

Rebuttal Figure 2

Therefore, we have revised the interpretation from “Collectively, these data show that endogenous TRIM5 α has the ability to prevent HIV-1 gene expression from reactivation by selected LRAs” to “Collectively, these data show that endogenous TRIM5 α has the ability to prevent HIV-1 gene expression under basal conditions or in the presence of selected LRAs”. Please see Lines 156-157, Page 8.

5. Figure 3

Switched GFP for Luc – not clear why – so essentially similar data as presented in Figures 1-2.

What is the basis for switching between a stable LTR suppression assay as in Figures 1-2, and a transient LTR expression as employed here. In Figure 3, cotransfection of TRIM5 and LTR-HIV constructs mean that the regulation at the LTR is likely to be wholly distinct to the regulation measured in the chronically infected scenario. Some comment on this would be valuable.

Response: We thank the reviewer for the helpful comment. We agree with the reviewer that the regulation of transfection of TRIM5 on LTR-HIV constructs is not fully the same as the regulation measured in the chronically infected scenario. As the HIV-1 LTR promoter serves as the main platform where transcription factors, transcription repressors, and epigenetic regulatory factors contribute to HIV latency, we first examined whether TRIM5 could also impact the HIV-1 LTR by

overexpressing or depleting TRIM5 in the presence of episomal LTR-HIV reporter constructs. With the implication obtained from these transfected LTR-HIV construct assays, we could then turn back to the chronically infected cells for certain ChIP–qPCR assays to explore the detailed mechanisms by which TRIM5 affects HIV latency in the chronically infected scenario.

We have now added a comment on the basis for switching from a stable LTR suppression assay to a transient LTR expression assay: “It has been shown that transcriptional silencing of HIV-1 replication is associated with multiple layers of regulation, including inadequate availability of transcription factors, recruitment of transcription repressors, and epigenetic modifications at the HIV-1 LTR. We then set out to examine whether TRIM5 α exerts a direct effect on the HIV-1 LTR by employing an LTR-driven firefly luciferase reporter assay”. Please see Lines 160-164, Page 9.

3c – Interesting result. How is TRIM5 only able to antagonise TNF-induced transcription and not Tat driven elongation, when the latter requires transcription initiation and production of the TAR element to which Tat binds (in complex with CDK9 and Cyclin T1)? What is the logic for this experiment?

Response: We thank the reviewer for the helpful comment. One possibility to explain “TRIM5 only able to antagonise TNF-induced transcription and not Tat driven elongation” is that TRIM5 may be counteracted by Tat. One study reported that SAMHD1 was able to impair TNF α -induced and PMA-induced HIV-1 reactivation but had no effect on the HIV-1 LTR in the presence of Tat, with an uncharacterized mechanism⁷. Another study reported that the HIV-1 Tat protein interacts with CPSF-73 and counteracts its repressive activity on the HIV-1 LTR promoter⁸. Tat has been reported to disassociate HDAC1 from HIV-1 LTR⁹. Thus, it is possible that Tat abrogates TRIM5 α -mediated suppression of the HIV-1 LTR by counterregulating HDAC1 recruitment to the HIV-1 LTR.

In this study, we dissected the mechanism by which TRIM5 α suppresses the activation of HIV-1 LTR by TNF α once we did not observe the effect of TRIM5 α on

Tat-activated LTR-driven expression. We will seek the mechanism by which Tat counteracts TRIM5 α -mediated suppression of HIV-1 LTR activation in our next research project.

3d – we can't see the values in untreated conditions. Log scale would be clearer. How were these graphs normalised (if at all)?

Response: We thank the reviewer for pointing this out. We apologize for the unclarity of this graph. We have now separated the panel to obtain a better view of the values of each bar. Please see Revised Figure 3d.

Revised Figure 3d

The luciferase activity of the reporters was normalized by the luciferase activity of RL-TK (*Renilla luciferase*), which was transfected along in each of the reporter assays, to normalize the transfection efficiency. We apologize for the unclarity. We have now added a description of the method of normalization to the text in Lines 164-166, Page 9: “The luciferase activity of the reporters was normalized to the luciferase activity of RL-TK (*Renilla luciferase*).”.

3e – empty plasmid control is missing. How were these values normalised to one another?

Response: We thank the reviewer for the helpful comment. Our original data of Figure 3e indeed included the empty plasmid control as well as rhesus TRIM5 α ,

shown below (Rebuttal Figure 3), so now we have added back the data of empty plasmid control in the Revised Figure 3e.

Rebuttal Figure 3

The luciferase activity of the reporters was normalized by the luciferase activity of RL-TK (*Renilla luciferase*), which was transfected along in each of the reporter assays, to normalize the transfection efficiency. We apologize for the unclearness. We have now added a description of the method of normalization to the text in Lines 164-166, Page 9: “The luciferase activity of the reporters was normalized to the luciferase activity of RL-TK (*Renilla luciferase*).”.

3c-3e – why are the RLU units so different between these experiments?

Response: We thank the reviewer for the comment. A possible reason for the difference in RLU in Figure 3c-3e is that the luciferase activity of Full-LTR is much higher than that of core-LTR. We conducted an experiment that directly compared the luciferase activity of these two constructs, and the results are shown below (Rebuttal letter Figure 4). In addition, the transfection amounts of reporter plasmids and RL-TK may vary between independent experiments, which leads to fluctuations in the final relative luciferase activity units.

Rebuttal Figure 4

In 3h-i, there seems to be a TNF-independent TRIM5 phenotype. In 3m-n, there seems to be a TNF-dependent TRIM5 phenotype. Would the authors comment on this? Both assays involved TRIM5 depletion.

Response: We thank the reviewer for pointing this out. We agree with the reviewer that there is a TNF-independent TRIM5 phenotype in Figure 3h-i, which is not consistent with the TNF-dependent TRIM5 phenotype in 3 m-n. We conducted the Figure 3h-i experiments two more times, and the results are shown below (Rebuttal letter Figure 5). The results showed that TRIM5 depletion by KD had little impact on basal LTR activity. Only when TNF was present could TRIM5 KD lead to the activation of the LTR. The new data are included in Revised Figure 3h, i.

Rebuttal Figure 5

Revised Figure 3h, i

The authors do not entertain the possibility of clonal artefact. Either multiple KO clones should be compared, or 1-2 clones should be reconstituted with TRIM5 to restore the parental phenotype. Otherwise, clone 10# could just be moderately more sensitive to TNF than the parental line.

Response: We thank the reviewer for the helpful comment. We actually obtained 2 clones with clear TRIM5 KO effect, which we named 10# and 16#, please see Revised Figure 3k. In the previous experiments, we only utilized 10# for the functional assays. As the reviewer suggested, we have now examined the effect of both TRIM5-KO 10# and 16# on HIV-1 LTR-driven gene expression. The new results showed that both clones yielded similar phenotypes in enhancing the TNF-induced activities of LTRs. Please see Revised Figure 3j-m.

Revised Figure 3j-n

6. Figure 4

Same concern as above re: clonal effect.

Response: We thank the reviewer for the helpful comment. As the reviewer suggested, we examined the effect of both TRIM5-KO 10# and 16# on NF- κ B- and Sp1-driven gene expression. The new results showed that both clones significantly potentiated the TNF-induced activities of NF- κ B-luc and Sp1-luc. Please see Revised Figure 4e, f.

Revised Figure 4e, f

Page 11, line 201, TRIM5 activates the Δ SP1 core LTR presumably because TRIM5 stimulates NF κ B expression when transfected into 293T cells.

Response: We thank the reviewer for the helpful comment. We completely agree with the reviewer that TRIM5 activates the Δ SP1 core LTR presumably because TRIM5 stimulates NF κ B expression. We have added this presumption to the sentence; please see Lines 202-205, Page 11: “in the absence of TNF α treatment, TRIM5 α suppressed the transcriptional activity of the Δ NF- κ B core LTR while activating the Δ Sp1 core LTR (Fig. 4a, b), presumably because TRIM5 α stimulates NF- κ B-luc activity”.

4c – we can't see the black bars or any change between conditions, either separate panels or a log panel should be used, although this might alter the conclusions made?

Response: We thank the reviewer for the helpful comment. As the reviewer suggested, we have separated the panel to obtain a better view of the values of each bar. The results showed that TRIM5 α was able to activate the 5* κ B reporter in the absence of a stimulus, which is consistent with previous reports^{10, 11} and our result from the Δ Sp1 core LTR reporter assay from Figure 4b. Therefore, the conclusions made are not altered. Please see Revised Figure 4b and 4c.

Revised Figure 4b, c

4g – Pg 12 line 224 – it's not possible to draw this conclusion from the data presented, the p65 blot is too dirty to see the NF κ B band in the nuclear fractions.

Response: We thank the reviewer for the helpful comment. We have reconducted this same experiment and obtained clearer western blots. The western

blots of our reconducted experiment clearly showed that p65 nuclear translocation in TRIM5 α -KD J-Lat cells decreased compared to that in control cells in the absence of TNF α but showed no difference between TRIM5 α -KD cells and control cells in the presence of TNF α . Please see Revised Figure 4g.

Revised Figure 4g

4h – p-IkBa blot too dirty to draw conclusion.

Response: We thank the reviewer for the helpful comment. We have reconducted this same experiment and obtained clearer western blots. The western blots of our reconducted experiment clearly showed that overexpression or knockout of TRIM5 α had little effect on the TNF α -mediated activation of I κ B α . Please see Revised Figure 4i.

Revised Figure 4i

7. Figure 5

It should be noted that TRIM5 itself stimulates an NFkB signal in 293T cells. The phenotype of the mutants used is inversely correlated with their NFkB activity (via TRIM5 E3 activity). Is this a coincidence or relevant to the phenotype?

Response: We thank the reviewer for the helpful comment. The luciferase experiments shown in Figure 5c were performed in the presence of TNF α . Therefore, the Δ RING mutant losing the TRIM5 α -WT-mediated suppression of TNF-induced NFkB activity may result from the loss of RING-mediated recruitment of HDAC1 to the p50 mechanism.

8. Figure 6

Generally the blots are overexposed making their interpretation difficult.

For the majority of panels in this figure there is a lack of an IgG control antibody that is important to know that the IPs are specific. Otherwise, the mirroring of input with IP could simply be non-specific binding of proteins to the protein A agarose etc.

Response: We thank the reviewer for the helpful comment. We apologize for not including the IgG control to distinguish nonspecific binding from specific binding. Therefore, we reperfomed the Co-IPs with the IgG control to confirm the TRIM5 α -mediated promotion of the interaction between HDAC1 and Sp1 as well as between HDAC1 and p50. The new results showed that there were no nonspecific bands in the IgG control. Please see Revised Figure 6a-c.

Revised Figure 6a-c

6f – pg14 line 274 – in fact, it looks like the effect of TRIM5 on HDAC1-sp1 binding is

weaker in the presence of TNF, and not equivalent as suggested in the text.

Response: We thank the reviewer very much for the constructive comment. We agree with the reviewer that the effect of TRIM5 α on the enhancement of the HDAC1-sp1 interaction is weaker in the presence of TNF α than in the absence of TNF α . We calculated the grayscale values of the HDAC1-Myc bands and compared the TRIM5 α -mediated HDAC1-sp1 interaction with or without TNF α ; please see Revised Figure 6f. Meanwhile, we found that the effect of TRIM5 α on the HDAC1-p50 interaction is weaker in the presence of TNF α , as shown in Revised Figure 6g.

Revised Figure 6f, g

Thus, these data indicate that TNF treatment attenuates the promoting effect of TRIM5 on the HDAC1-sp1 interaction and the HDAC1-p50 interaction. To further elucidate the implication of the TNF α -induced effect on TRIM5 α , we examined the recruitment of TRIM5 α to the HIV-1 LTR in the absence or presence of TNF α by ChIP-qPCR analysis. The results showed that TRIM5 α was disassociated from the LTR upon TNF α treatment, as shown in Revised Figure 8d and Supplementary Figure 4. Altogether, these data may explain why TRIM5 α suppressed the activity of HIV-1 LTR in a TNF α -dependent manner.

Revised Figure 8d

Revised Supplementary Figure 4

Thus, we have revised the description from “We also showed that the TRIM5 α -mediated HDAC1–Sp1 and HDAC1–p50 interactions persisted in the presence of TNF α (Fig. 6f, g)” to “Notably, the results showed that the TRIM5 α -promoted HDAC1–Sp1 and HDAC1–p50 interactions were attenuated by treatment with TNF α (Fig. 6f, g) and that the association of TRIM5 α and Sp1 was also markedly reduced upon TNF α treatment (Fig. 6f), suggesting the release of TRIM5 α -mediated suppression of Sp1 and NF- κ B under latency-reversing stimulation”. Please see Lines 281-285, Page 15. We also discussed these results in the Discussion section: “Further studies revealed that TNF α treatment released TRIM5 α from the HIV-1 LTR and disrupted TRIM5 α ’s effect on the promotion of the HDAC1–Sp1 and HDAC1–p50 interactions (Fig. 8d, Supplementary Fig. 4 and Fig. 6f, g), which may explain why the effect of TRIM5 α on Sp1-luc and NF- κ B-luc was exaggerated upon TNF α treatment”, please see Line 436-440, Page 22.

9. Figure 7

7j-m – this data appear to be misinterpreted. TRIM5 KO does not rescue the inhibition caused by HDAC1 expression on LTR transcription. Rather, the fold inhibition caused by HDAC1 is equivalent in control or TRIM5 KD cells. In 7 m there

seems to be negligible effect caused by HDAC1.

Response: We thank the reviewer for the helpful comment. We agree with the reviewer that in Figure 7 j-m, TRIM5 KO does not rescue the inhibition caused by HDAC1 expression on LTR transcription, and that in 7 m there seems to be negligible effect caused by HDAC1. We repeated the experiments and obtained similar results. The fold inhibition by HDAC1 on LTR was calculated and compared between TRIM5 KD cells and control cells in Supplementary Figure 5c, d. Thus, the results indicated that TRIM5 is not fully responsible for the inhibitory effect of HDAC1 on LTR transcription. As HDAC1 has been reported to be regulated by a number of transcription factors, such as YY1, LSF, c-Myc and CTIP2^{9,12,13}, it is possible that HDAC1 retains efficient inhibition of HIV transcription through those factors.

Revised Supplementary Figure 5c, d

We have rewritten the description as follows: “However, TRIM5α depletion did not fully abolish the effect of HDAC1 on the repression of HIV-1 full/core LTR-driven transcription (Supplementary Fig. 5b-d) or the repression of HIV-1 latency activation by HDAC1 in J-Lat cells (Supplementary Fig. 5e, f), indicating that HDAC1 inhibition of LTR transcription is partly dependent on TRIM5α.”; please see Lines 347-351, Page 18.

Minor comments:

10.Pg 7 line 129 and line 137 – 1g, not 1e?

Response: We thank the reviewer for pointing this out. We have corrected it. Please see Line 130 and Line 138, Page 7.

11. Page 4 line 60 – the sentence is a little ambiguous. The TRIM5 RING is widely known to be an adaptor type E3 for ubiquitination, but not Sumoylation or ISGylation (as in, biochemical evidence, not just overexpressed proteins in cells). The RING fold in general however is involved in all 3 of these. So just to clarify whether this sentence relates to the RING fold in general, or specifically the TRIM5 RING.

Response: We thank the reviewer very much for pointing out this vagueness. This sentence is specifically related to TRIM5 RING. Thus, we changed the previous “The RING domain confers E3 ligase activity mediating ubiquitination, ISGylation, or SUMOylation of specific substrates and is also implicated in promoting the efficiency of higher order assembly together with the B-box 2 domain” to “The RING domain of TRIM5 α confers E3 ligase activity mediating ubiquitination and is also implicated in promoting the efficiency of higher order assembly together with the B-box 2 domain”. Please see Lines 60-62, Page 4.

12. ‘PRY-SPRY’ not just ‘SPRY’ – e.g. Page 3 line 59

Response: We thank the reviewer for pointing this out. We have corrected ‘SPRY’ to ‘PRY-SPRY’. Please see Line 58, Page 3.

References

1. Ciccosanti F, *et al.* High Levels of TRIM5 α Are Associated with Xenophagy in HIV-1-Infected Long-Term Nonprogressors. *Cells* **10**, (2021).
2. Li X, Sodroski J. The TRIM5 α B-box 2 domain promotes cooperative binding to the retroviral capsid by mediating higher-order self-association. *J Virol* **82**, 11495-11502 (2008).
3. Wagner JM, *et al.* Mechanism of B-box 2 domain-mediated higher-order assembly of the retroviral restriction factor TRIM5 α . *Elife* **5**, e16309 (2016).

4. Li Z, Guo J, Wu Y, Zhou Q. The BET bromodomain inhibitor JQ1 activates HIV latency through antagonizing Brd4 inhibition of Tat-transactivation. *Nucleic Acids Res* **41**, 277-287 (2013).
5. Contreras X, *et al.* Suberoylanilide hydroxamic acid reactivates HIV from latently infected cells. *J Biol Chem* **284**, 6782-6789 (2009).
6. Bartholomeeusen K, Fujinaga K, Xiang Y, Peterlin BM. Histone Deacetylase Inhibitors (HDACis) That Release the Positive Transcription Elongation Factor b (P-TEFb) from Its Inhibitory Complex Also Activate HIV Transcription. *J Biol Chem* **288**, 14400-14407 (2013).
7. Antonucci JM, *et al.* SAMHD1 Impairs HIV-1 Gene Expression and Negatively Modulates Reactivation of Viral Latency in CD4+ T Cells. *J Virol* **92**, e00292-00218 (2018).
8. de la Vega L, Sanchez-Duffhues G, Fresno M, Schmitz ML, Munoz E, Calzado MA. The 73 kDa subunit of the CPSF complex binds to the HIV-1 LTR promoter and functions as a negative regulatory factor that is inhibited by the HIV-1 Tat protein. *J Mol Biol* **372**, 317-330 (2007).
9. He G, Margolis DM. Counterregulation of Chromatin Deacetylation and Histone Deacetylase Occupancy at the Integrated Promoter of Human Immunodeficiency Virus Type 1 (HIV-1) by the HIV-1 Repressor YY1 and HIV-1 Activator Tat. *Mol Cell Biol* **22**, 2965-2973 (2002).
10. Pertel T, *et al.* TRIM5 is an innate immune sensor for the retrovirus capsid lattice. *Nature* **472**, 361-365 (2011).
11. Zhu JW, Mu D, Liu FL, Luo MT, Luo RH, Zheng YT. Activation of NF-kappaB induced by TRIMCyp showing a discrepancy between owl monkey and northern pig-tailed macaque. *Mol Immunol* **101**, 627-634 (2018).
12. Jiang G, Espeseth A, Hazuda DJ, Margolis DM. c-Myc and Sp1 Contribute to Proviral Latency by Recruiting Histone Deacetylase 1 to the Human Immunodeficiency Virus Type 1 Promoter. *J Virol* **81**, 10914-10923 (2007).
13. Marban C, *et al.* Recruitment of chromatin-modifying enzymes by CTIP2 promotes HIV-1 transcriptional silencing. *EMBO J* **26**, 412-423 (2007).

REVIEWERS' COMMENTS

Reviewer #1 (Remarks to the Author):

All my concerns have been addressed in a satisfactory manner. I will be happy to support the acceptance of this current study.

Reviewer #2 (Remarks to the Author):

Thanks for the attention to all comments. I recommend this article to be accepted for publication.